# Nucleosomal DNA has topological memory

Joana Segura [1,3], Ofelia Díaz-Ingelmo [1], Belén Martínez-García [1], Alba Ayats-Fraile [1], Christoforos Nikolaou [2] & Joaquim Roca [1] ✉

One elusive aspect of the chromosome architecture is how it constrains the DNA topology. Nucleosomes stabilise negative DNA supercoils by restraining a DNA linking number difference (ΔLk) of about −1.26. However, whether this capacity is uniform across the genome is unknown. Here, we calculate the ΔLk restrained by over 4000 nucleosomes in yeast cells. To achieve this, we insert each nucleosome in a circular minichromosome and perform Topo-seq, a high-throughput procedure to inspect the topology of circular DNA libraries in one gel electrophoresis. We show that nucleosomes inherently restrain distinct ΔLk values depending on their genomic origin. Nucleosome DNA topologies differ at gene bodies (ΔLk = −1.29), intergenic regions (ΔLk = −1.23), rDNA genes (ΔLk = −1.24) and telomeric regions (ΔLk = −1.07). Nucleosomes near the transcription start and termination sites also exhibit singular DNA topologies. Our findings demonstrate that nucleosome DNA topology is imprinted by its native chromatin context and persists when the nucleosome is relocated.

High-throughput analyses increasingly improve our knowledge of chromatin structure and function genome-wide[1,2]. However, a fundamental trait that remains elusive to current technologies is the topology of the chromatinized DNA. We do not know how the DNA double helix is twisted and bent along chromatin. We define "constrained topolome" as DNA deformations stabilized by chromatin, and "unconstrained topolome" as those resulting from the mechanical stress the DNA undergoes during genome activities[3–5].

The principal actor of the constrained topolome in eukaryotic cells is the nucleosome, the DNA packaging unit of chromatin, in which about 147 base pairs (bp) of DNA make nearly 1.6 left-handed super-helical turns around a histone octamer[6]. However, this canonical configuration is neither uniform nor static[7–9]. The DNA nucleotide sequence, histone composition and modifications affect nucleosome conformation and stability[10–12]. So far, numerous studies have mapped nucleosome structure and position in model organisms, such as budding yeast[13–17]. However, no study has yet addressed how nucleosomes constrain the topology of DNA throughout the genome.

In a previous study, we developed a strategy to measure the DNA linking number difference (ΔLk) restrained by nucleosomes in vivo[18]. We compared by DNA electrophoresis the ΔLk constrained by yeast circular minichromosomes before and after inserting an additional nucleosome. By doing so, we found that not all nucleosomes restrain ΔLk (DNA supercoils) to the same extent, being the average value of

ΔLk = −1.26[18]. Our next goal was to determine the ΔLk constrained by all individual nucleosomes. However, since it was unfeasible to perform thousands of electrophoretic analyses (one for each nucleosome), we needed another strategy to circumvent this problem.

Here, we develop a high-throughput procedure termed "Topo-seq" to analyse the topology of large libraries of circular DNA molecules. We apply Topo-seq to a library of 4000 nucleosomes hosted in yeast circular minichromosomes in vivo. Our results reveal that nucleosomes inherently restrain distinctive ΔLk values depending on their genomic origin. Therefore, nucleosome DNA topology is an intrinsic trait imprinted by the native chromatin context, which persists when allocated somewhere else. We discuss the implications of this nucleosomal feature and the potential of Topo-seq to disclose further DNA topology genome-wide.

## Results

### Construction of minichromosomes hosting a nucleosome DNA library

As in our previous study[18], we digested budding yeast chromatin with Micrococcal nuclease and purified a pool of mono-nucleosomal DNA fragments (≈150 bp) that combined increasing digestion rates (Fig. 1a). After repairing the DNA ends, we added adaptors to insert the nucleosome DNA library in the YCp1.3 (1341 bp) yeast circular minichromosome (Supplementary Fig. 1). Next, we introduced these

[1]DNA Topology Lab, Molecular Biology Institute of Barcelona (IBMB-CSIC), Barcelona, Spain. [2]Computational Genomics Group, BSRC Alexander Fleming, Athens, Greece. [3]Present address: Centro de Biología Molecular Severo Ochoa (CSIC/UAM), Madrid, Spain. ✉e-mail: joaquim.roca@ibmb.csic.es

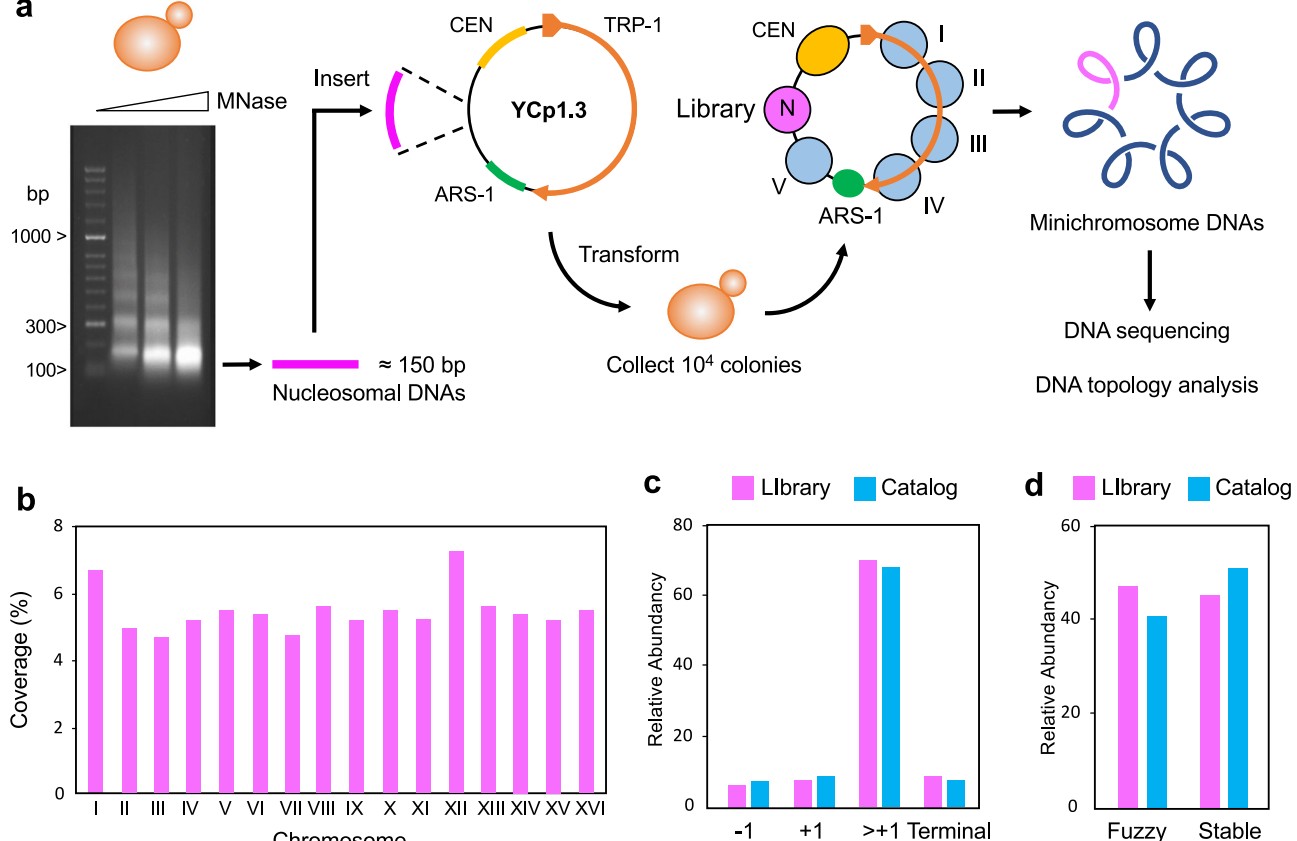

**Fig. 1 | Mono-nucleosome DNA library construction. a** The gel shows yeast chromatin digested with increasing amounts of Micrococcal nuclease. Nucleosome DNA fragments of about 150 bp (pink) generated in a single experiment with increasing degrees of digestion were pooled and inserted into the YCp1.3 minichromosome DNA (1341 bp) to transform yeast cells. About 10⁴ transformants were collected and pooled for sequencing and topology analysis of the minichromosome DNAs. The allocation of the nucleosome DNA library (N) into the chromatin structure of YCp1.3 is illustrated (see Supplementary Fig. 1 for details). **b** Coverage (%) of the nucleosome DNA library (see Supplementary Data 1 for nucleosome coordinates) across the 16 yeast chromosomes (I-XVI) normalized per kb. **c** Relative abundance of nucleosomes from different positions relative to the transcription start site (−1, +1 > +1) and the transcription terminal site. **d** Relative abundance of fuzzy and stable nucleosomes. In (**c**) and (**d**), the nucleosome DNA library ($n = 4276$, pink) is compared to the full catalogue of yeast nucleosomes ($n = 61110$, blue). Source data are provided as a Source data file.

minichromosomes, each containing one nucleosome of the library, into yeast cells and obtained about ten thousand transformants. We pooled all these colonies and extracted their DNA (Fig. 1a). Parallel DNA sequencing of the minichromosomes revealed a library of 8369 mono-nucleosome DNA fragments of length $144 \pm 21$ bp (mean ± SD). Nearly all (> 91%) of these DNA fragments overlapped (> 50 bp) with the genomic coordinates of 4276 previously referenced nucleosomes[13] (Supplementary Data 1, Supplementary Fig. 2). The chromosomal distribution (Fig. 1b), genomic type (Fig. 1c), stability (Fig. 1d) and positioning (Supplementary Fig. 3) of the identified nucleosomes were comparable to that of the bulk of referenced nucleosomes[13], signifying the nucleosome DNA library was representative.

### Conceptualization leading to Topo-seq

Due to DNA thermodynamics, covalently closed circular DNA molecules present a Gaussian distribution of DNA linking number (Lk) topoisomers, visualized as a ladder of DNA bands by agarose gel electrophoresis[19]. When the mean Lk value of a DNA molecule relaxed in vitro ($Lk^0$) is compared to that of the DNA in a circular minichromosome ($Lk^{Chr}$), the Lk difference is the ΔLk constrained by the minichromosome chromatin ($Lk^0 - Lk^{Chr} = \Delta Lk^{Chr}$). Upon adding a new nucleosome to such minichromosome, $\Delta Lk^{Chr}$ changes to $\Delta Lk^{Chr+nuc}$ and the resulting difference ($\Delta Lk^{Chr+nuc} - \Delta Lk^{Chr} = \Delta Lk^{nuc}$) equals the ΔLk restrained by the new nucleosome (Supplementary Fig. 4). Following

this strategy, we previously showed that most nucleosomes restrain ΔLk values of about −1.26 in vivo[18]. However, this strategy was unfeasible for determining the ΔLk of each nucleosome of our library, as it would require running thousands of electrophoresis. To circumvent this problem, we ran instead all minichromosome DNAs containing the library in a single electrophoresis lane (Fig. 2a, lane 1). As a result, the pool of Lk distributions overlapped and produced a smear rather than a ladder of Lk topoisomers. This smear occurred because the nucleosome sequences of the library were not of equal length ($144 \pm 21$ bp) and, since these length differences were not multiples of the helical repeat of DNA (10.5 bp), most ladders of Lk topoisomers had different phasing. We evidenced this misalignment by running, in the same gel, the Lk distributions of individual minichromosomes that carried nucleosome DNA sequences differing by only 1 to 5 bp in length (Fig. 2a, lanes 2−7). Note that Lk distributions can have equal or distinct Lk mean (red lines) irrespectively of whether their Lk topoisomers are aligned (Fig. 2b, lanes 2−7). This gel electrophoresis also evidenced that the mean Lk of the pool of Lk distributions produced by the nucleosome DNA library (Fig. 2b, lane 1) virtually matched with that produced by individual nucleosomes that restrained ΔLk −1.26 (Fig. 2b, lanes 2−5). As in our previous study[18], this observation denoted that most fragments of the DNA library assembled nucleosomes and that the average ΔLk constrained by them is −1.26. Importantly, this value differs markedly from the average ΔLk of −0.84 constrained by an analogous library of non-nucleosomal (prokaryotic) DNA fragments

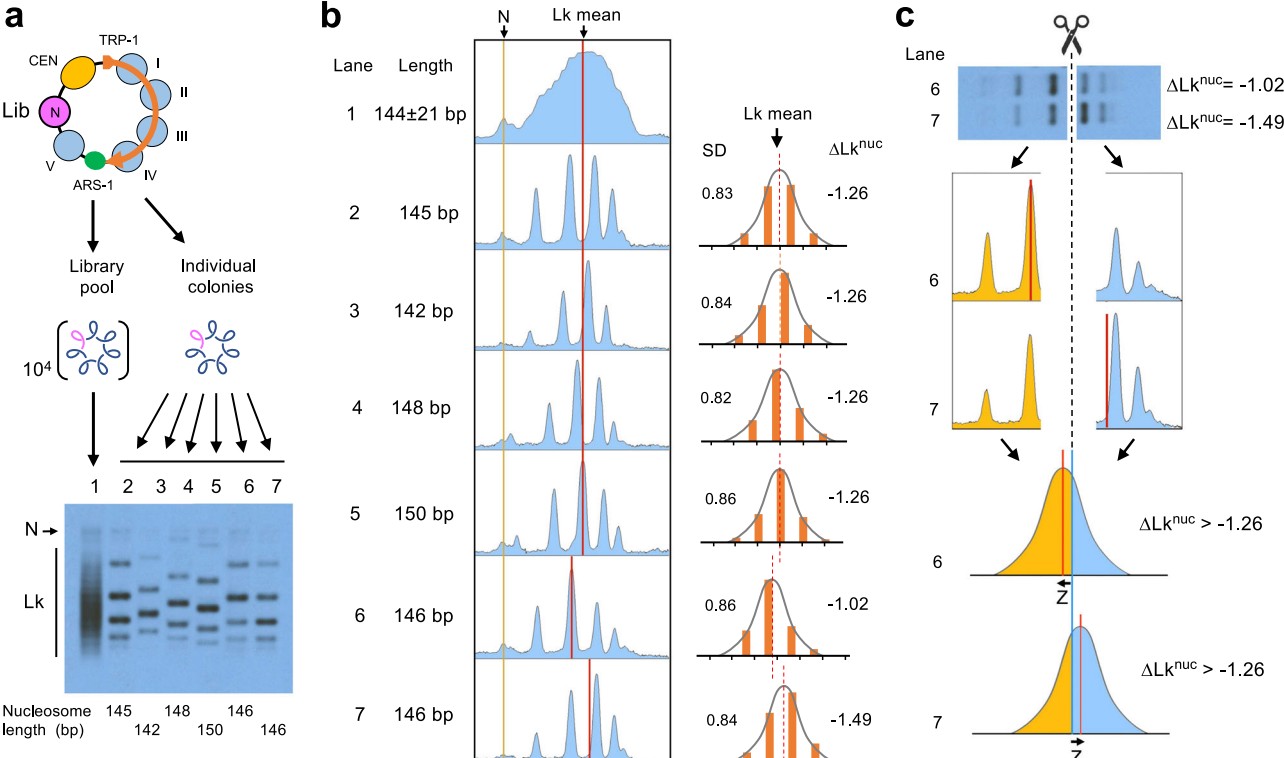

**Fig. 2 | Topo-seq conceptualization. a** Gel electrophoresis of Lk distributions. Lane 1, pool of $10^4$ minichromosome DNAs hosting the nucleosome DNA library. Lanes 2–7, individual minichromosome DNAs hosting a nucleosome with the indicated DNA length (bp). Electrophoresis conditions are described in Methods. N, nicked circles. Lk, topoisomers. **b** Gel densitometry (left, in blue) and relative topoisomer intensities (right, in orange) of the previous Lk distributions (lanes 2–7), indicating their Lk mean position (red lines), standard deviation (SD) and the ΔLk restrained by the hosted nucleosome ($\Delta Lk^{nuc}$) calculated as described in Supplementary Fig 4. **c** Cutting the Lk distributions of sample 6 ($\Delta Lk^{nuc} = -1.02$) and 7 ($\Delta Lk^{nuc} = -1.49$) at the global Lk mean position ($\Delta Lk^{nuc} = -1.26$) produces unequal topoisomer DNA abundances at each side of the cut (orange and blue). As Lk distributions are Gaussian, the distinct partition probabilities of DNA abundance translate into Z-scores (Z) of a normal probability distribution. These Z-scores reflect how far the Lk mean of each distribution is from that of the global average (the cut site), where $\Delta Lk^{nuc} = -1.26$. Source data are provided as a Source data file.

(Supplementary Fig. 5), corroborating thereby that the average ΔLk of −1.26 is specific to nucleosomal DNA.

At this point, we realized that within the library pool of Lk distributions (Fig. 2b, lane 1), those from nucleosomes restraining ΔLk < −1.26 and >−1.26 should respectively have their Lk mean slightly ahead and behind the global average (for example, samples 6 and 7 in Fig. 2b). Accordingly, upon cutting the smear of Lk distributions at the level of the global mean, nucleosome DNA sequences restraining ΔLk <−1.26 and >−1.26 would differently enrich at each side of the cut (Fig. 2c, top). The partition probability of each nucleosome DNA sequence would reflect how far its $\Delta Lk^{nuc}$ deviates from the global mean ($\Delta Lk^{nuc} = -1.26$, at the cut site); and, since Lk distributions are Gaussian, this deviation would correlate to the Z-score of a normal distribution (Fig. 2c, bottom). In this way, we could classify all nucleosomes of the library by their capacity to restrain ΔLk. We termed this procedure "Topo-seq" since it relies on sequencing DNA topoisomers eluted from different sections of an electrophoresis gel.

**Topo-seq and calculation of the ΔLk restrained by nucleosomes**

We cut into two sections the gel slab comprising the overlapping Lk distributions of our nucleosome DNA library (Fig. 3a), such that the top section (A) enriched nucleosome DNA sequences restraining ΔLk > −1.26 and the bottom section (B) those restraining ΔLk <−1.26. We eluted the DNA from both sections, amplified the nucleosome DNAs by PCR and sequenced them. In this step, note that since $\Delta Lk^{nuc}$ calculations derive from the partition ratio of each sequence in these gel sections, potential biases due to the length and bp composition of nucleosomal DNAs would similarly occur on both sides and thus have

little effect on the partition ratios. As expected, all DNA sequences of the nucleosome DNA library were present in both sections but with distinct partition probabilities (Fig. 3b). Section B was enriched in short nucleosomal DNA sequences (< 140 bp) since the Lk distributions of the corresponding DNA circles migrated faster during electrophoresis. However, the partition ratios A/B varied up to a factor of 10 within each length group, which denoted a diversity in nucleosome DNA topology (Fig. 3c).

To calculate the ΔLk restrained by each nucleosome, we converted the partition probability of each nucleosome DNA sequence into a Z-score of a normal distribution of mean zero and a standard deviation one. Next, we multiplied these Z-scores by 0.85, which is the standard deviation of the Lk distributions of the ≈1.5 kb minichromosome DNAs hosting the nucleosome DNA library (Fig. 2b). The resulting values indicated how far (ΔLk units) the mean of each Lk distribution was from the global Lk average ($\Delta Lk^{nuc}$ of −1.26 at the cut site) (Fig. 3d). Next, we adjusted these ΔLk scores by taking into account that the nucleosome DNA sequences had different lengths. Firstly, to correct the effect of misaligned Lk ladders (Supplementary Fig. 6), we subtracted from each ΔLk score the average ΔLk of all the sequences of the same length (i.e., aligned ladders). These corrected ΔLk values now reflected how the topology of each nucleosome compared to those of equal length (Fig. 3e). Secondly, we calculated that the overall electrophoretic velocity of the Lk distributions shifted the equivalent of 0.007 Lk units for each bp difference in length (Supplementary Fig. 7). Therefore, we adjusted the ΔLk values by adding or subtracting 0.007 for each bp difference from 144 bp. Finally, as the current ΔLk values were relative to the global average

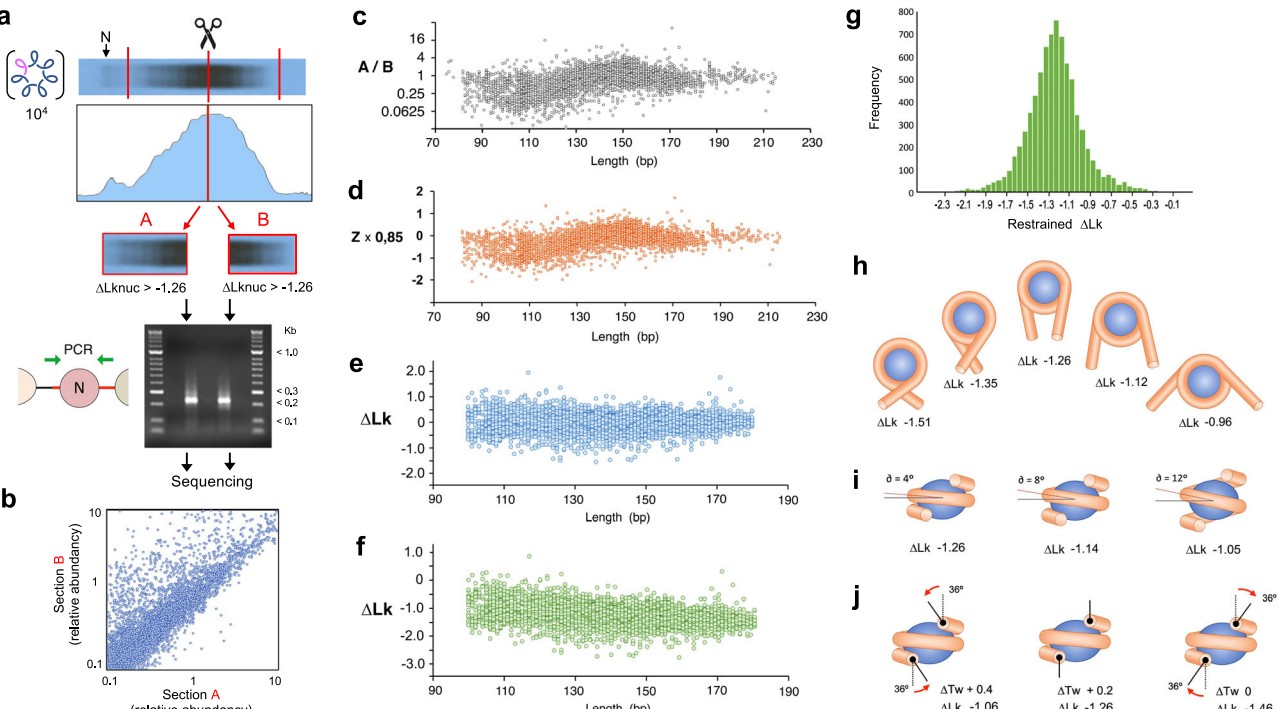

**Fig. 3 | Calculation of the ΔLk restrained by individual nucleosomes via Topo-seq. a** Splitting of the Lk distributions of lane 1 in Fig. 2a at the level of the global Lk mean ($\Delta Lk^{nuc} = -1.26$); and PCR products of the nucleosome sequences eluted from section A and B. **b** Plot of the abundance of individual nucleosome DNA sequences in sections A and B. **c** Plot of the partition probability of each nucleosome DNA sequence (ratio A/B) vs the sequence length (bp). **d** Plot of the partition probabilities converted into Z-scores of a normal distribution and multiplied by 0.85 (SD of Lk distributions). **e** Corrected ΔLk scores after subtracting the average ΔLk of all the sequences of the same length. **f** Plot of ΔLk values after adding or subtracting 0.007 units for each bp difference from 144 bp and finally adding −1.26. See

Supplementary Data 2 for calculated values. **g** Histogram of ΔLk values restrained by the nucleosome DNA library (mean = −1.26, SD = 0.33). Analyses in (**a**–**f**) were conducted once with the nucleosome library ($n = 8369$) described in Fig. 1.
**h** Models of restrained ΔLk as a function of the number of DNA super-helical turns ($N$), considering $\Delta Tw = +0.2$ and $\Delta Wr = N(1-\sin 4°)$. **i** Models of restrained ΔLk as a function of the pitch angle of DNA super-helical turns ($\partial$), considering $\Delta Tw = +0.2$ and $\Delta Wr = 1.56(1-\sin \partial)$. **j** Twisting angle (red arrows) at the entry and exit of DNA to produce ΔTw of 0, +0.2, +0.4 and the indicated ΔLk, considering $\Delta Wr = 1.56(1-\sin 4°)$. See Supplementary Fig. 10 for detailed correlations of ΔWr and ΔTw on ΔLk. Source data are provided as a Source data file.

($\Delta Lk^{nuc} = -1.26$), we added −1.26 to obtain the actual ΔLk restrained by each nucleosome (Fig. 3f, Supplementary Data 2).

### Conversion of ΔLk$^{nuc}$ into twist and writhe deformations of DNA

The ΔLk$^{nuc}$ values calculated via Topo-seq (mean −1.26, SD 0.33) denoted a substantial heterogeneity in the nucleosome DNA topology (Fig. 3g). This heterogeneity did not correlate with the extent of overlap of our library with previously referenced nucleosomes (Supplementary Fig. 8); and it was neither consequent to variability in DNA linker length, which we tested by examining the ΔLk$^{nuc}$ of a nucleosome with different linker sizes (Supplementary Fig. 9 and Supplementary Table 1). Therefore, since $\Delta Lk = \Delta Tw + \Delta Wr$, nucleosomes with distinct ΔLk$^{nuc}$ values could differ in restraining ΔTw (helical twist) but, more likely, by their capacity to constrain ΔWr (super-helical turns). We then modelled how ΔWr values would translate into ΔLk by considering $\Delta Tw = +0.2$ (as in the canonic nucleosome) and $\Delta Wr = N(1-\sin \partial)$[18], where N is the number of wrapped super-helical turns (Fig. 3h, Supplementary Fig. 10a) and $\partial$ their pitch angle (Fig. 3i, Supplementary Fig. 10b). We also considered how ΔTw values would translate into ΔLk, being $\Delta Wr = -1.46$ (Fig. 3j, Supplementary Fig. 10c). For simplicity, we depicted only symmetric shapes, but asymmetric changes of ΔWr and ΔTw are likely to occur and combine producing a broad spectrum of possible conformations even for nucleosomes restraining similar ΔLk values.

### Correlation of ΔLk$^{nuc}$ with the composition of nucleosome DNAs

We examined whether the nucleosome capacity to restrain ΔLk would depend on the DNA nucleotide composition, which affects DNA

flexibility and curvature[20]. We found that GC content was significantly higher ($p < 2e{-}12$) for nucleosomes restraining $\Delta Lk > -1.26$ (GC 42.3%) compared to those of $\Delta Lk < -1.26$ (GC 40.7%) (Fig. 4a). Regarding dinucleotide frequencies, nucleosomal DNAs of the library were enriched in AA/TT/TA dinucleotides, being CG/GG/GC dinucleotides the less abundant. However, this general trend faded in the nucleosomes restraining less negative ΔLk values, which exhibited an abundance of other dinucleotides such as CA/TG (Fig. 4b). As expected, AA/TT/TA dinucleotides had a periodical spacing close to the DNA helical repeat (10−11 nts), though nucleosomes restraining less negative ΔLk values also presented other dinucleotides with short periodic patterns (Fig. 4c). Since DNA composition affects the position stability of nucleosomes[21,22], we examined whether the restrained ΔLk correlated to nucleosome stability. We found that the positional fuzziness of nucleosomes, measured in their natural loci[13], had no relation with their intrinsic capacity to constrain ΔLk (Fig. 4d).

### Correlation of ΔLk$^{nuc}$ with the genomic origin of nucleosomes

Next, we examined whether the nucleosome capacity to restrain ΔLk would depend on their genomic origin. In this case, several unexpected correlations emerged (Fig. 5, Supplementary Data 3). Gene body nucleosomes constrained a mean ΔLk$^{nuc}$ of −1.29, which was distinct ($p < 0.001$) from the mean ΔLk$^{nuc}$ of −1.23 constrained by intergenic nucleosomes (Fig. 5a). The ΔLk restrained by gene body nucleosomes (transcribed by Pol-II) was also different ($p < 0.004$) from the mean ΔLk$^{nuc}$ of −1.24 restrained by nucleosomes of rDNA genes (transcribed by Pol-I and -III) (Fig. 5a). Within gene body nucleosomes, there were also significant differences depending on the nucleosome

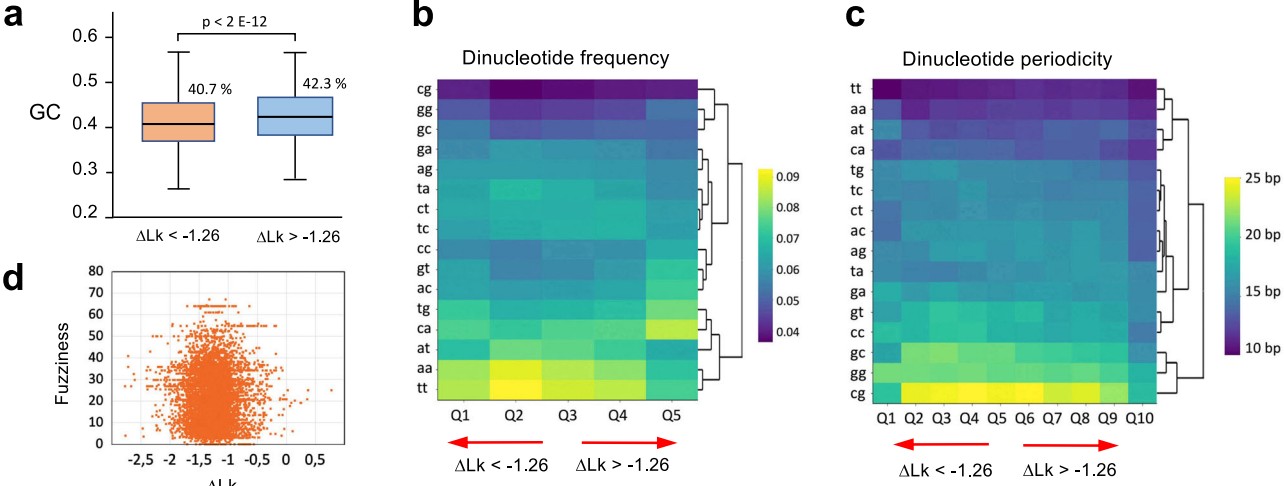

**Fig. 4 | ΔLk correlation with the DNA base pair composition of nucleosomes.**
**a** GC content of nucleosomal DNAs of the library with ΔLk$^{nuc}$ values <−1.26 (*n* = 4187) and >−1.26 (*n* = 3611). In the box plots, the centre line denotes the median value, while the box marks the 25th (upper limit) to 75th (lower limit) percentiles of the dataset. The black whiskers mark the 5th and 95th percentiles. A *p*-value of 1.8 E

−12 was determined by a two-sided Wilcoxon test. **b** Heatmap of dinucleotide frequencies for ΔLk$^{nuc}$ values of the library (*n* = 8369) in 5 quantiles. **c** Heatmap of dinucleotide periodicities for ΔLk$^{nuc}$ values of the library (*n* = 8369) in 10 quantiles. **d** Scatter plot of ΔLk$^{nuc}$ values of the library (*n* = 8369) against the native positional fuzziness of nucleosomes. Source data are provided as a Source data file.

position relative to the transcription start site (TSS) and termination site (TTS) (Fig. 5b). Nucleosomes at position −1 from TSS restrained a mean ΔLk$^{nuc}$ of −1.27, a value more similar to gene body than intergenic nucleosomes. Nucleosomes at position +1, stably positioned downstream of the TSS, also restrained a mean ΔLk$^{nuc}$ of −1.27, like the bulk of gene body nucleosomes. However, two gene body nucleosomes constrained ΔLk values significantly more negative than the others. Nucleosomes at position +2 restrained a mean ΔLk$^{nuc}$ of −1.33 (*p* < 0.001), and terminal nucleosomes at the TTS restrained a mean ΔLk$^{nuc}$ of −1.32 (*p* < 0.008). Lastly, Topo-seq uncovered that the nucleosomes from telomeric regions[23] were those with the most atypical DNA topology. These nucleosomes presented short DNA sequences (136 bp on average) and a mean ΔLk$^{nuc}$ of −1.07, highly deviated (*p* < E−60) from the mean ΔLk$^{nuc}$ of −1.27 of non-telomeric nucleosomes (Fig. 5c). The topology of telomeric nucleosomes was significantly distinct even when compared to short nucleosomal DNAs of non-telomeric regions (Supplementary Fig. 11).

### Validation of the ΔLk$^{nuc}$ dependence on the origin of nucleosomes

To validate the location-dependent differences of ΔLk$^{nuc}$ uncovered by Topo-seq, we measured the ΔLk$^{nuc}$ of individual nucleosomes of each genomic region. We chose 18 nucleosomes (3 gene boby, 3 intergene, 3 rDNA, 3 telomeric, 3 nuc+2, 3 terminal), whose ΔLk$^{nuc}$ value determined via Topo-seq was representative of the mean ΔLk$^{nuc}$ of their corresponding region (Supplementary Table 1). We amplified and inserted these sequences (147 bp in length) in the YCp1.3 plasmid, transformed yeast cells and examined the individual Lk distributions of the resulting minichromosomes (Supplementary Fig. 12). The ΔLk$^{nuc}$ values of these 18 nucleosomes presented a good correlation (R$^2$ ≈ 0.9) with the ΔLk$^{nuc}$ values calculated via Topo-seq (Fig. 5d). These results confirmed that the intrinsic capacity of nucleosomes to restrain DNA supercoils depends on their genomic origin; and proved that Topo-seq is a faithful high-throughput procedure to examine the topology of circular DNA libraries.

### Discussion

Our study provides two advances toward the unravelling of intracellular DNA topology. First, it shows the feasibility of Topo-seq, a high-throughput procedure for examining topolomes. Second, it

demonstrates that DNA retains a topological memory of its deformation by chromatin.

To date, gel electrophoresis of circular DNA molecules is the only procedure to quantitatively characterize changes in the linking number (supercoiling) and the degree of entanglement (knotting or catenation) of DNA. Therefore, to examine local DNA topology within cellular chromosomes, one strategy has been popping out DNA rings at specific chromatin regions[24–26]. However, this procedure would be extremely tedious, if not impracticable, for conducting genome-wide analyses of DNA topology. Topo-seq relies on the sequencing of DNA topoisomers eluted from distinct sections of one gel electrophoresis and, therefore, permits inspecting the topology of multiple DNA molecules at once. We show that DNA amounts as little as 0.1 ng, comprising a library of thousands of DNA constructs, can be eluted from an agarose gel and processed for high-throughput sequencing. Since the calculation of ΔLk values derives from the partition probability of each sequence in different gel sections, these ratios can be calculated regardless of the abundance of each sequence in the initial library pool. In the present study, since the circular DNAs containing the nucleosome DNA library were of similar length (about 1.5 kb), we resolved the pool of ΔLk distributions in a one-dimensional gel and then cut it into two sections at the level of the global Lk mean. We then applied corrections for the little length differences, which effect on DNA migration was always smaller than the gel distance between individual Lk topoisomers. However, for future developments of Topo-seq, DNA libraries containing constructs of multiple lengths (1 to 15 kb) could be analysed by cutting the gel into many small sections such that the position and abundance of any DNA topoisomer could be determined irrespective of its size. Another possibility is running the DNA library in a two-dimensional gel. In this case, Lk distributions would resolve into a diagonal of arched ladders, such that Lk topoisomers from one DNA length do not overlay with those of another.

After calculating the ΔLk values constrained by our nucleosome DNA library, we found structural and functional correlations that validated the reliability of Topo-seq. Importantly, since we interrogated the nucleosomes outside their native genomic locus, the restrained ΔLk values had to rely on the intrinsic traits of their DNA sequences. In this respect, since high GC content makes the DNA stiffer[27,28], the increased GC content of nucleosomes restraining less negative ΔLk values might reflect a reduced capacity of DNA to

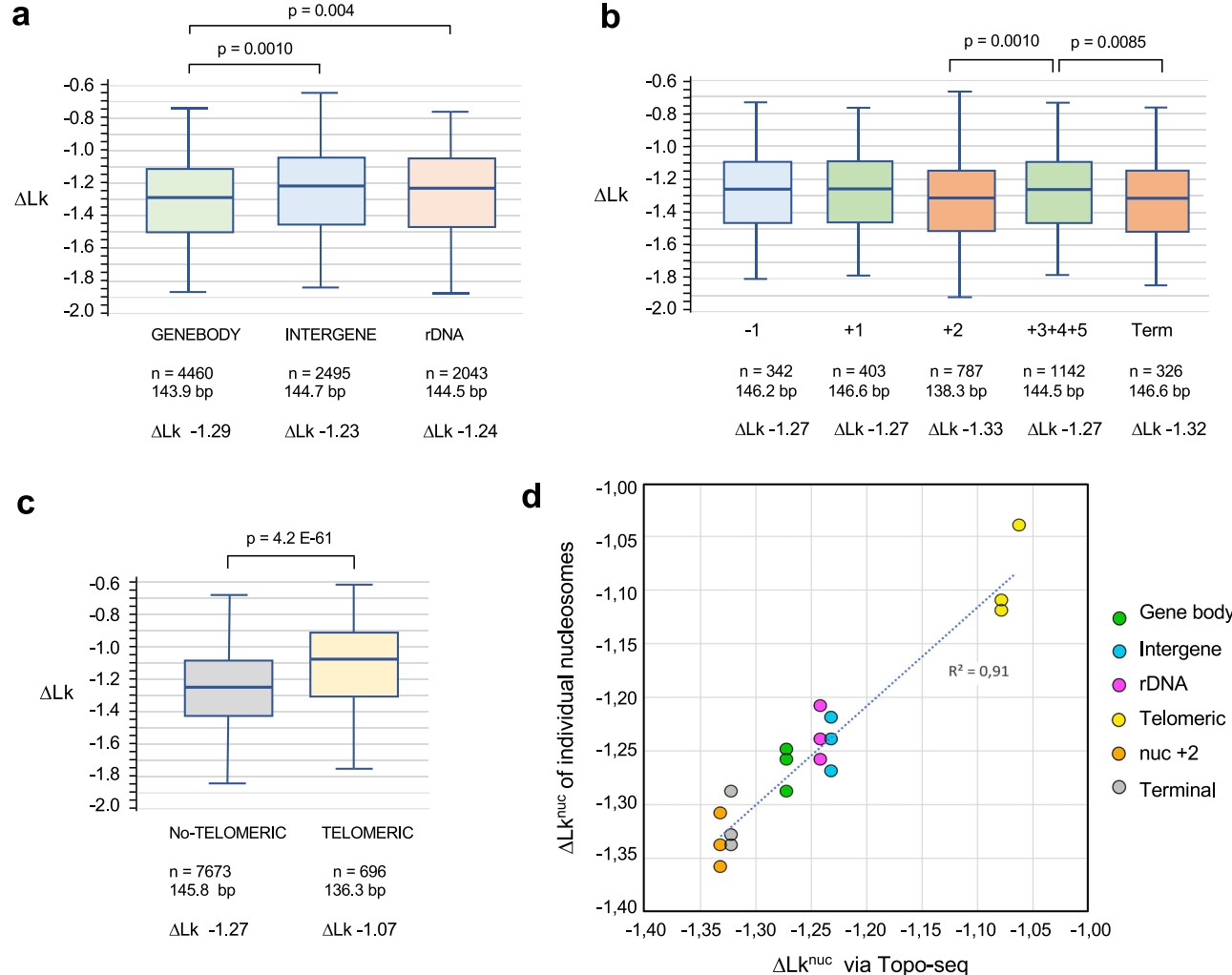

**Fig. 5 | ΔLk correlation with the native genomic allocation of nucleosomes.**
**a** ΔLk restrained by gene body (*n* = 4460), intergenic (*n* = 2495) and rDNA nucleosomes (*n* = 2043). **b** ΔLk restrained nucleosomes at positions −1 (*n* = 342), +1 (*n* = 403), +2 (*n* = 787), +3 to +5 (*n* = 1142) relative to the TSS and the TTS (Term). **c** ΔLk restrained by non-telomeric (*n* = 7673) and telomeric nucleosomes (*n* = 696). In (**a**–**c**) box plots, the centre line denotes the median value, while the box marks the 25th (upper limit) to 75th (lower limit) percentiles of the dataset. The black whiskers mark the 5th and 95th percentiles. In (**a**–**b**), *p*-values were determined by two-sided ANOVA with Tukey's test. In (**c**), the *p*-value was determined by an

unpaired two-sided *t*-test. The bottom rows in (**a**–**c**) indicate the number (*n*), average sequence length (bp) and mean ΔLk of nucleosomes in each class. The ΔLk$^{nuc}$ and genomic allocation of individual nucleosomes are described in Supplementary Data 3. **d** Correlation of ΔLk$^{nuc}$ values obtained via Topo-seq with those calculated via the analyses of individual Lk distributions (see Supplementary Fig 12). The scatter plot shows the Lk$^{nuc}$ of 18 representative nucleosomes (3 gene body, 3 intergenic, 3 rDNA, 3 telomeric, 3 nuc+2, 3 terminal) obtained by both procedures. Nucleosome coordinates and ΔLk$^{nuc}$ values are specified in Supplementary Table 1. Source data are provided as a Source data file.

completely or stably wrap around histone cores. As expected, the nucleosome DNA library exhibited a 10–11 bp periodicity of AA/TT/TA dinucleotides, which is known to favour nucleosome positioning and stability[21,22]. But, curiously, this dinucleotide pattern did not correlate with the capacity to restrain ΔLk. Only the nucleosomes constraining less negative ΔLk values presented atypical dinucleotide periodicities, which emerged from the repetitive DNA sequences of telomeric regions, as discussed below. Therefore, nucleosome DNA topology might not only depend on dinucleotide parameters, which are also weak predictors of the intrinsic cyclability of DNA[29]. Moreover, the no correlation between the nucleosome capacity to restrain ΔLk and their native genomic stability also denoted these are independent structural traits. Conceivably, positional stability depends on the central turn of DNA around the H3-H4 tetramer; and DNA topology mainly relies on interactions with the H2A-H2B dimers configuring the geometry of nucleosome entry and exit DNA segments.

In contrast to the nucleotide composition, the genomic origin of nucleosomes presented striking correlations to their intrinsic capacity

to restrain ΔLk (Fig. 6). The gene body nucleosomes (mean ΔLk$^{nuc}$ of −1.29) are distinct from the intergenic ones (mean ΔLk$^{nuc}$ of −1.23). If this difference relies on the extent of DNA wrapping (ΔWr) and the subsequent different orientations of DNA entry and exit segments, nucleosome arrays in genic and intergenic regions must adopt distinct architectures. Interestingly, this inference might relate to the distinctive folding motifs described in yeast chromatin, where the gene body nucleosomes tend to present tetrahedron folds and the intergenic ones rhomboidal folds[16] (Fig. 6). The distinct DNA topology of gene body and intergenic nucleosomes could also reflect that the firsts are suited to confront the torque generated during DNA transcription[30] and to facilitate DNA tearing by transcribing RNA polymerases[31]. However, the topology of gene body nucleosome is not observed in the rDNA genes (mean ΔLk$^{nuc}$ of −1.24). Since rDNA nucleosomes are evicted by the transcribing trains of RNA polymerase I[32,33], their distinctive DNA topology probably reflects this unique biophysical context, quite different from genes transcribed by RNA polymerase II.

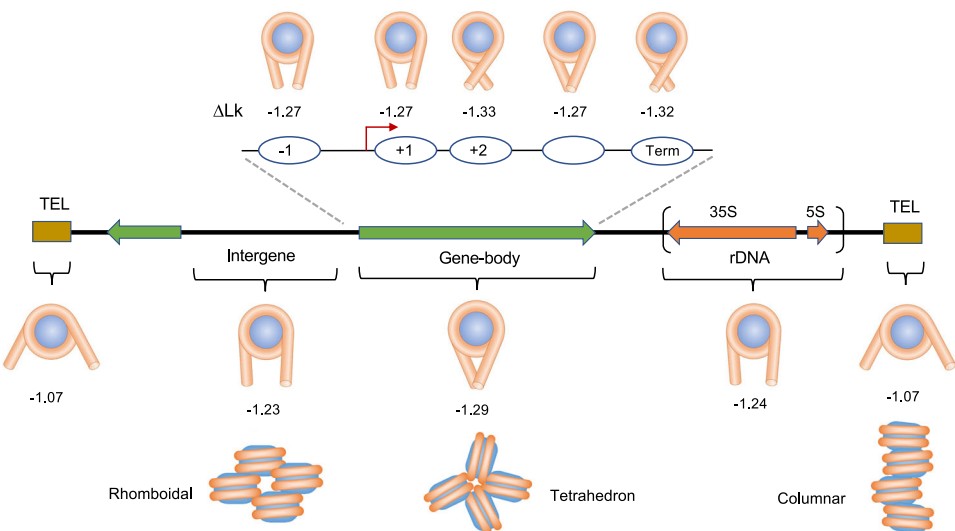

**Fig. 6 | Models of nucleosomal DNA topology depending on their genomic origin.** The $\Delta Lk^{nuc}$ values of nucleosomes are interpreted and modelled as different capacities to wrap the DNA (brown) around a histone core (blue). Tetra- nucleosomes illustrate chromatin folding motifs proposed for gene bodies (tetrahedron), intergenic regions (rhomboidal) and telomeric chromatin (columnar).

Within gene body nucleosomes, we expected nucleosomes at positions −1 and +1 relative to the TSS would have singular DNA topologies since they delimit a nucleosome-free region (NFR) in most gene promoters[17,34,35]. Their position relies on transcription factors and chromatin remodelling activities that push them out of the NFR[36–39]. The +1 nucleosomes also interact with the transcription preinitiation complex (PIC)[40,41]. However, the intrinsic capacity of the −1 and +1 nucleosomes to restrain $\Delta Lk$ is similar to that of most gene body ones. In contrast, Topo-seq uncovered singular DNA topologies for nucleosomes at positions +2 and the TTS. The case of the +2 nucleosomes ($\Delta Lk^{nuc} = -1.33$) is striking because they presented DNA sequences shorter (138,3 bp) than the global average and still restrained more negative $\Delta Lk$ values than any other nucleosome. The +1 and +2 nucleosomes have high positional stability[35] and are recognized as a dinucleosome by remodelling complexes[36,42]. This feature and a plausible role in the promoter-proximal pausing of RNA Polymerase II[43,44] might explain the singular topology of +2 nucleosomes. Regarding the TTS nucleosomes, their high capacity to restrain DNA supercoils ($\Delta Lk^{nuc} = -1.32$) likely relates to their role as transcription elongation barriers. Interestingly, their capacity to mark transcription termination depends on a proper configuration of their DNA entry-exit site and, thus, their DNA topology[45].

Lastly, Topo-seq uncovered nucleosomes from telomeric regions[23] as those with the most unusual DNA topology ($\Delta Lk^{nuc} = -1.07$). Remarkably, these nucleosomes presented shorter DNA sequences (1363 bp) than the non-telomeric ones; and exhibited unusual patterns of dinucleotides since they host abundant short DNA sequence repeats[46]. Therefore, their reduced capacity to restrain $\Delta Lk$ likely reflects an incomplete or distinctive wrapping of the DNA (Fig. 6). Their peculiar DNA topology might relate to the recently reported columnar architecture of telomeric chromatin[47] that shows nucleosomes stacked on each other with a repeat length of about 132 bp, letting the DNA wound as a continuous superhelix (Fig. 6).

Taken together, the correlations of $\Delta Lk^{nuc}$ with the genomic origin of nucleosomes demonstrate that some aspects of the native chromatin context are imprinted on the topology of nucleosomal DNA, such that they reverberate when nucleosomes are placed outside their natural genomic loci. In the same way that dinucleotide periodicity has been optimized for wrapping nucleosomal DNA around histone cores, other traits of the nucleosomal DNA sequences might have been adjusted to adapt nucleosomes to specific chromatin configurations or genomic environments. Therefore, since the intrinsic topology of nucleosomal DNA retains a memory of its native allocation, this inherent trait of nucleosomes becomes a new determinant of chromatin architecture. However, note that the $\Delta Lk^{nuc}$ values measured in the minichromosome hosting the nucleosome DNA library are not necessarily equal to the $\Delta Lk^{nuc}$ of the nucleosomes in their native allocation, where each nucleosome interplays with its corresponding chromatin environment. Likewise, note that the $\Delta Lk^{nuc}$ values of some nucleosomal DNA sequences could be also altered by the recruitment of additional DNA binding factors, nucleosome remodelling activities or reflect partial nucleosome occupancy. We foresee that applying Topo-seq to more extensive libraries of chromatin elements, in combination with chromatin immunoprecipitation or other biochemical selectors of chromatin composition, will provide unprecedented insights into the interplay of chromatin and DNA topology genome-wide. Ultimately, the need for circular minichromosomes to host these libraries could be dispensable if, for instance, the bulk of intracellular chromatin is fragmented and circularised in situ. In this case, upon running an entire genomic pool of DNA rings in a single gel electrophoresis, Topo-seq could readily disclose the topolome of any cell type.

## Methods

### Construction of the nucleosome DNA library

*Saccharomyces cerevisiae* FY251 cells (*MATa his3-Δ200 leu2-D1 trp1-Δ63 ura3–52*) were grown at 28 °C in 250 ml of rich medium until OD 1.0. Cells were collected, washed with water, and incubated with 80 ml of 1 M Sorbitol, 30 mM DTT for 15 min at 28 °C. Next, 625 U of Lyticase (Sigma-Aldrich L2524) and 10 μL of 4 M NaOH were added to the cell's suspension, and the incubation continued until >80% of cells converted into spheroplasts. Spheroplasts were washed with 1 M Sorbitol and resuspended in 1.5 ml of hypotonic lysis buffer (1 mM $CaCl_2$ 5 mM $KH_2PO_4$ 1 mM PMSF) at 24 °C. After adding 30 units of micrococcal nuclease (Sigma-Aldrich N3755), the lysate was incubated at 24 °C. Aliquots of 300 μl were quenched with 20 mM EDTA 1% SDS at different incubation times (3 to 30 min). Gel electrophoresis of digested chromatin was done in 1% agarose in TBE buffer, at 80 V for 3 h. Mononucleosome DNA fragments (about 150 bp in length) produced at successive digestion times were gel-eluted and pooled. The DNA ends produced by micrococcal nuclease were repaired by removing terminal 3′-phosphates with T4-polynucleotide kinase and filled with

Klenow and T4-DNA polymerase activities. The resulting A-tailed DNA fragments were ligated to T-tailed adaptors that provided degenerated *Asc1* or *BamH1* ligation ends (see Supplementary Fig. 1). The adapted DNA fragments were inserted between the *Asc1* and *BamH1* sites of the YCp1.3 DNA (1341 bp), which was hosted in a bacterial plasmid. After amplifying these plasmids in *E. coli* cells, they were digested with *Not1* to release DNA segments of about 1.55 kb that comprised YCp1.3 with the inserted library of mono-nucleosome DNAs (≈ 200 bp). These DNA segments were gel purified and circularised by ligating their *Not1* ends.

### Extraction of minichromosome DNAs hosting the library

Monomeric circles of YCp1.3 hosting the nucleosome DNA library were gel-purified and used to transform FY251 via electroporation. About ten thousand colonies were collected from agar plates containing yeast synthetic media (TRP dropout). These colonies were pooled, washed with water, resuspended in 200 ml of TRP dropout and incubated at 28 °C for 2 h. The cells were then fixed by mixing the suspension with one cold volume (−20 °C) of ET solution (Ethanol 95%, Toluene 28 mM, Tris HCl pH 8.8 20 mM, EDTA 5 mM). The cells were sedimented at room temperature, washed twice with water, resuspended in 400 μl of TE, and transferred to a 1.5-ml microfuge tube containing 400 μl of phenol and 400 μl of acid-washed glass beads (425–600 μm, Sigma). Mechanic lysis of >80% cells was achieved by shaking the tubes in a FastPrep® apparatus for 10 s at power 5. The aqueous phase of the lysate was collected, extracted with chloroform, precipitated with ethanol, and dissolved in 100 μl of TE containing RNAse-A. After a 15-min incubation at 37 °C, the samples were extracted with phenol and chloroform, the DNA precipitated with ethanol and dissolved in 50 μl of TE.

### Construction of minichromosomes hosting individual nucleosomes

Individual nucleosomal DNA sequences of specific length were obtained from yeast genomic DNA via PCR amplification (NEB Taq Polymerase) by using the primers described in Supplementary Table 1. The resulting A-tailed DNA fragments were ligated to T-tailed adaptors to be inserted between the *Asc1* and *BamH1* sites of the YCp1.3 DNA, as described above. Yeast cells transformed with minichromosomes hosting the individual nucleosomes were grown in 20 ml of TRP dropout at 28 °C. When the cultures reached the exponential phase (OD ≈ 1), the cells were fixed and the minichromosome DNAs were extracted as described above.

### Electrophoresis of *Lk* distributions

The DNA of minichromosomes that hosted individual nucleosomes or the nucleosome DNA library was loaded onto 1.4% (w/v) agarose gels. Electrophoreses were carried out at 2.5 V/cm for 18 h in TBE buffer (89 mM Tris-borate and 2 mM EDTA) containing 0.55 μg/ml chloroquine. Gels were blot-transferred to a nylon membrane and probed at 60 °C with the YCp1.3 DNA labelled with AlkPhos Direct (GE Healthcare®). Chemiluminescent signals of increasing exposure periods were recorded on X-ray films. Non-saturated signals of individual Lk topoisomers and bins of pooled Lk distributions were quantified with ImageJ. The Lk mean of the Lk distributions was determined as previously described[18] and illustrated in Supplementary Fig. 4.

### Topo-seq

About 500 ng of yeast DNA, including that of minichromosomes hosting the nucleosome DNA library (about 0.5 ng), were loaded in two adjacent lanes of a gel containing 1.4% (w/v) agarose (Ultrapure grade, nzytech® MB05202) and electrophoresed at 2.5 V/cm for 18 h in TBE buffer. The gel slab of the first lane was kept at 4 °C in TBE, while the second lane was blot-transferred and probed to determine the position of the Lk mean of the overlapping Lk distributions. The gel slab of the first lane was then cut at the level of this Lk mean in two sections (A and

B) of about 25 × 10 × 4 mm each, such that section A contained the top half of the overlapping Lk distributions; and section B, the bottom half. DNAs from both gel sections were eluted and captured using the Geneclean II kit®. DNA fragments of the nucleosome DNA library present in each section were amplified via PCR (NEB Taq Polymerase) by using as primers the sequences of the *Asc1* and *BamH1* adaptors described in Supplementary Fig. 1. The obtained amplicons of about 200 bp were sequenced in duplicate on two Flowcell Lane Index units (Illumina MiSeq v2 2×150 pb, paired-end reads) and the resulting FASTQ data subjected to QC using Cutadapt (1.12). The reads from two Flowcell lanes (sequencing replicates 1 and 2) were combined into a unique analysis dataset. The library sequences were then mapped to the *Saccharomyces cerevisiae* reference genome (R64-1-1 from Ensembl Genomes) using bowtie (v1.1.2). Once nucleosome coordinates were established, alignment and size statistics were calculated using SAM-Tools and Picard. Subsequent analyses were performed by integrating published nucleosome data sets[13,48] and using BEDTools (v2.27) and Galaxy. The calculation of ΔLk values using the Topo-seq DNA sequencing data is described in the results section and Supplementary Data 2. This process includes cunning the relative abundances of nucleosomal DNA sequences in the gel sections A and B, the conversion of these partition probabilities into Z scores of a normal distribution, the transformation of Z scores into Lk values, and the adjustment of these ΔLk values to take into account the length differences within the library of nucleosomal DNA sequences.

### ΔLk modelling of nucleosomes

To model restrained ΔLk values as a function of ΔWr, the value of ΔTw was fixed to +0.2 and restrained ΔWr values were calculated from ΔWr = N(1–sin ∂). The number of wrapped super-helical turns (N) was depicted as the length of the arcs in contact with the cylindrical histone core, leaving the entry and exit segments of DNA as detached tangents and with a pitch angle (∂) of 4°. As super-helical turns are left-handed, N < 0. The pitch angle (∂) of the super-helical turns was calculated from the height at the entry and exit points of 1.56 super-helical turns. To model the restrained ΔLk as a function of ΔTw, the nucleosome ΔWr was fixed to −1.46 (= −1.56 (1–sin 4°)) and restrained ΔTw values of 0 and +0.4 were illustrated as the twisting angle observed at the entry and exit segments of DNA relative to ΔTw in the cannon nucleosome (ΔTw = +0.2).

### ΔLk correlation with DNA composition and nucleosome origin

Sequence properties of nucleosome DNAs including GC content, dinucleotide frequency and periodicity were analysed in R using the seqinR package. Mean dinucleotide frequencies were calculated for nucleosome ΔLk values split into five quantiles. Dinucleotide periodicities were measured as the mean distance (in bps) of consecutive dinucleotides of the same type and calculated for nucleosome ΔLk values split into ten quantiles. The native allocation of identified nucleosomes within a gene body, intergenic region and rDNA genes was defined by Jiang and Pugh (2009). Positional fuzziness of nucleosomes was determined as the standard deviation in bp of the nucleosome position from the 6 datasets compiled by Jian and Puhg[13]. Nucleosomes of telomeric regions were as defined in The *Saccharomyces* Genome Database (SGD)[23]. Statistic tests for the significance of ΔLk correlations are described in the figure legends.

### Statistics and reproducibility

Statistical tests were performed using Microsoft Excel version 16.7, GraphPad Prism version 8.0, and IBMSPSS Statistics version 19.0. The statistical test used for the data shown in each figure is noted in the corresponding figure legend, and significant statistical differences are noted as *p*-values. When indicated, values are reported as mean values ± standard deviation. No data were excluded from the analyses. The Topo-seq analysis was conducted once using the described library of

nucleosomal DNAs. The reproducibility of the library results was validated by examining individual representative nucleosomes.

### Reporting summary

Further information on research design is available in the Nature Portfolio Reporting Summary linked to this article.

## Data availability

The data generated in this study are available with this manuscript and its Supplementary Information. Source data have been deposited in NCBI's Gene Expression Omnibus[49] and are accessible through GEO Series accession number GSE228623. Source data are provided with this paper.

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

## Acknowledgements

The authors thank the CNAG Sequencing Service and the CRG Bioinformatics Core Facility for assistance; F. Azorín, G. Vicent and V. Zhurkin for discussing results. This work is supported by the Plan Estatal de Investigación Científica y Técnica of Spain, with grant PID2019-109482GB-I00 to J.R.; research fellowships BES-2012-061167 to J.S. and PRE2020-093378 to A.A-F.

## Author contributions

J.R. conceived Topo-seq and supervised the study. J.S. designed and performed the Topo-seq experiments. C.N. and J.S. designed and performed the bioinformatics analyses. O.D-I., B.M-G. and A.A-F. performed validation experiments. J.R. produced nucleosome models and wrote the manuscript.

## Competing interests

The authors declare no competing interests.
