## [Peer Review File · Nature Communications]

Nucleosomal DNA has topological memoryREVIEWER COMMENTS

Reviewer #1 (Remarks to the Author):

Segura et al. report that DNA associated with nucleosomes restrain different ΔLk (linking number difference) depending on their genomic localization. This work follows up their previous report of a ΔLk of -1.26 for the bulk of nucleosomes in *S. cerevisiae*. Briefly, the authors cloned mononucleosomal DNA in a plasmid (Figure 1) and measure the ΔLk after its introduction in *S. cerevisiae* either in a few individual molecules (Figure 2) or in two subsets of the whole library fractionated by the differential mobility of the topoisomers (Figure 3). The work is potentially interesting to a wide readership but, in my opinion, the results are preliminary because several additional controls would be necessary to support the conclusion that nucleosomes confer a different ΔLk depending on their genomic localization (Figure 4).

1. To isolate mononucleosomal DNA, the authors digest chromatin with micrococcal nuclease (MNase) and recover the band of ~ 150 bp from a gel (Figure 1a). As a quality control of nucleosomal DNA, they estimate that $>91\%$ of the recovered fragments overlap at least 50 bp to previously reported nucleosomes. I think that a more rigorous control would be necessary, given that 50 bp is a third of nucleosomal DNA and many non-nucleosomal fragments 150 bp long would also overlap > 50 bp to nucleosomal DNA. The recovered MNase fragments should be sequenced and aligned to the *S. cerevisiae* reference genome to generate a nucleosome occupancy map. Its comparison to other good quality maps in the literature would guarantee the nucleosomal origin of the recovered fragments. This control is important to ensure a homogeneous coverage of the genome. It is very common that AT- or GC-rich fragments are over or underrepresented in the final dataset depending on a) the degree of digestion with MNase, b) biased enrichment caused by some kits during the preparation of the sequencing libraries and c) differential amplification during the PCR step.

2. To calculate the ΔLk of nucleosomes (ΔLk_{nuc}), the authors subtract the Lk of the empty plasmid (LkChr) (1341 bp) from the Lk of the plasmid harbouring nucleosomal inserts (LkChr+nuc) ($1341 + \sim 150$) (Extended data Fig. 2). This assumes that the difference in size between the plasmids will not affect their topology and, therefore, the ΔLk_{nuc} will be a direct consequence of the topological properties of the nucleosomal insert.

A possible control to clarify this point would be to generate in the same plasmid a library of random fragments ~ 150 bp derived from, for example, naked *S. cerevisiae* DNA partially digested with MNase or from DNA of prokaryotic origin having the same G+C content as *S. cerevisiae*. If these control libraries have the same ΔLk as the empty plasmid, it could be concluded that $\Delta Lk_{Chr+nuc}$ is really due to the nucleosomal inserts. However, if ΔLk of the control libraries is closer to $\Delta Lk_{Chr+nuc}$, it would be necessary to revise the conclusion that it was due exclusively to the properties of nucleosomal DNA.

3. The fragments analyzed are 144 ± 21 bp long (page 4). This means that the same 147 bp of nucleosomal DNA could have heterogeneous flanking sequences in the population of fragments due to variability of MNase digestion in different cells. It would be important to assess the effect of this heterogeneity in the ΔLk . A possible experiment to test it directly would be to generate several nucleosomal sequences 147 bp long from specific well positioned nucleosomes in the genome with flanks of different sizes. They should be cloned in the plasmid to test their Lk individually as in Figure 2 to measure directly the effect of the flanks on the same nucleosomal sequences. The result could be used to confirm the more indirect estimate of 0.007 Lk units for each bp difference in length obtained by subtracting the average ΔLk of all sequences of the same length from the ΔLk score of each individual sequence (page 6).

4. To fractionate the distribution of topoisomers at higher resolution, the gel in Figure 3a could have been sliced in several fractions, not only in two. Increasing the resolution of the fractionation is very important because the relative representation of each fragment on each gel fraction is used to assign a specific Lk to each fragment (Suppl Table 3).

5. Differences in the mean ΔLk of nucleosomes on different genome locations (Figure 4) are very subtle and, given the concerns raised in the previous points, they should be experimentally confirmed. For example, some nucleosomal sequences from these different regions could be cloned in the same 1341 bp plasmid and their ΔLk should be determined individually to confirm the location dependent differences described in Figure 4.

Although the genome coverage of nucleosomal fragments seems to be uniform across chromosomes (Figure 1b), a possible concern in Figure 4 is that the proportion ribosomal DNA fragments (2043 out of 8998 = 22.7%) is more than 2-fold the approximately 10% of the genome represented by the rDNA genes. Also, the ΔLk of +1 nucleosomes is based on 403 nucleosomes only. Considering a conservative number of 5000 protein coding genes in *S. cerevisiae*, the analysis represent approximately 8% of all +1 nucleosomes.

6. As mentioned in point 1, small differences in the level of MNase digestion can cause significant variations on the quality and relative enrichment of nucleosomal DNA. For this reason, it should be tested how robust the differences reported in Figure 4 are to differences in the level of digestion with MNase. The Methods section indicates that mononucleosomal DNA fragments were eluted from the gel and pooled. Does pooling refer to the three lines of the gel in figure 1a? Would comparable results be obtained in duplicate experiments using mononucleosomal fragments from an independent experiment? The same reasoning applies to the cutting of the gel slab at the level of ΔLk of exactly -1.26 in the global distribution of topoisomers (Figure 3a). It seems that no replicate of this experiment was done. In other words, how robust are the results obtained to experimental variation?

Reviewer #2 (Remarks to the Author):

In this study, the authors introduce Topo-seq, a method to calculate linking number difference (dLk) introduced by ~ 150 bp fragments (genome-wide) into a reference plasmid. dLk is calculated usually using gel electrophoresis, but doing so for 1000s of possible sequences is almost impossible. Here, the authors elegantly demonstrate a sequencing based method that allows electrophoretic separation of a library of 1000s of plasmids with inserts corresponding to ~ 150 bp of DNA protected by nucleosomes in the yeast genome. They find +2 nucleosomes of genes and telomeric DNA to have significantly altered dLk compared to the rest of the genome, hinting at the intrinsic DNA sequences driving distinct chromatin structures in these regions. I have the following comments:

1. The authors do not discuss replicates. How reproducible is the dLk values from a genome-wide assay?

2. Related to point 1, the authors should compare the dLk values obtained from their assay to the ones determined by gel electrophoresis for a few nucleosomes (either from previous studies or select a few from this study). This will again validate the genome-wide dLk calculations. This is essential especially since the authors introduce a statistical approach with several steps to calculate the final dLk from their sequencing datasets and it has to be shown that their calculations yield accurate dLk values.

3. Although the authors mention dLk of "nucleosomes" throughout the text, we do not know if the sequences for nucleosomes on the plasmids. Similar to point 2, it would good to validate that insertion of a random 150 bp sequence that protected a nucleosome in the genome would result in the plasmid actually having an extra nucleosome.

4. The genome-wide analysis is very superficial. Some analyses that come to mind that haven't been

performed is, what is the correlation between transcription and dLk at +1, +2, and gene body? How does dLk change at transcription factor binding sites? How about centromeres? Is the in vivo linker length reflected by intrinsic dLk values? The authors need to think about how to make their genome-wide analysis more substantial.

Reviewer #3 (Remarks to the Author):

The authors present original research, which demonstrates that nucleosomal DNA topology is imprinted into DNA sequence. To achieve this, the authors placed a library of nucleosome-binding DNA fragments, obtained from MNase digestion, in the YCp1.3 yeast circular minichromosome. The library was amplified in yeast cells and run on gel electrophoresis. Due to varying size of library fragments, the authors obtained smeared distribution instead of clear separate bands expected for DNA topoisomers. The smearing prohibited direct quantification of topoisomers. To solve this problem the authors cleaved gel in two halves with an assumption that the mean of the smeared distribution corresponds to the average $\Delta(Lk) = -1.26$ per nucleosome (as measured in the previous publication of the same group), where Lk is a linking number. By sequencing minichromosomes extracted from both sides of the cleaved gel and counting the relative abundance of each library fragments on both sides of the gel, the authors built Z-score function to quantify $\Delta(Lk)$ of individual library fragments. The authors performed statistical analysis and showed that specific values of $\Delta(Lk)$ are enriched in different genomic regions (gene promoters, +/-1 nucleosome around TSS, gene bodies, etc.), which made authors to conclude that topology of the DNA wrapped around nucleosome has specific features that are encoded solely by DNA sequence and persists when relocated into different genomic context (i.e. into circular minichromosome).

While this is definitely an interesting study, it has a few problematic places in my opinion. It also lacks rigorous description of the method author used. This should be addressed before publishing.

1. I worry about potential bias from the fragment library size variability. The authors acknowledge that topoisomer's ladder is not aligned across multiple library fragments due to different fragment size. It could be a problem in my opinion if the shift in migration distance due to different fragment size is more than distance between two topoisomers. In this case you can get all topoisomers of a specific fragment on one half of the gel just because it migrates much slower or much faster due to the fragment size difference. At Fig. 2b the authors show how topoisomer peaks shift due to difference in the fragment size. They compare fragments in the range of 144-150 bp and see significant shift, which is however within acceptable range. However, the size of their library is quite broad 144 +/- 21 bp. Thus, it would be great to see how fragments randomly chosen from the library with a broad range of size distribution behave on gel. It would also be beneficial to see how the fragments picked from extremes of Z-scores value behave on gel. To reduce fragment library size bias the authors can try to restrict size of the fragments to the narrow window and sequence the library deeper to get enough statistics.

2. At Fig. 3 c-f, the authors partially try to assess the bias due to fragment library size variability. They write "Although section B slightly enriched in short nucleosome DNA sequences, the partition probabilities within each group denoted main differences in nucleosome DNA topology". It is hard to agree with term 'slightly' here. As an example, at Fig. 3c the difference in A/B ration varies by factor of 10 depending on fragment size. However, due to log scale this is hard to see. Also, to support statement that partition probabilities denote the main difference in DNA topology, the authors should pick up certain fragment length, randomly select representative number of fragments, do gels and compare to their partition function scores.

3. Normalization of Z-score by 0.85, which as authors wrote is the standard deviation of Lk distributions of the DNAs hosting nucleosome library, can be problematic since it doesn't take into account shift due to fragment size variability.

4. At Fig. 4h the authors try to interpret observed $\Delta(Lk)$ values in terms of the geometry of DNA wrapped around nucleosomes. While this is potentially possible interpretation, I'm wondering if the authors tried to consider other interpretations, which could be partial nucleosome occupancy of library fragments, or activity of remodeling complexes, which can change frequency of the observed topoisomeric states without changing DNA topology?

Minor comments:

1. The authors frequently use term 'nucleosome library' across the text, which is a bit misleading. I believe the authors should stick to 'nucleosome DNA library'.
2. Fig 1 legend: "In c and d, the nucleosome library (n=4276, pink) is compared to the full catalogue of the yeast nucleosomes." However, there are not pink color in those panels.
3. Page 6: "as in the cannon nucleosome". Should be canonic.

General response to the reviewer's comments

We very much thank the comments of the three reviewers, who acknowledge that our study presents original and elegant research that is potentially interesting to a wide readership. We are also grateful for their insightful remarks and suggestions, which allowed us to improve the quality of the work and strengthen the conclusions.

We appreciated that the three reviewers pointed out similar concerns about potential bias generated in our experimental approach and the need to validate the Topo-seq results by directly examining the ΔLk restrained by individual nucleosomes. Therefore, we have focussed the revision on these aspects. To this end, we incorporated more coauthors to complete the proposed experiments on time. In our point-by-point responses, we also explain why some of the suggested analyses cannot be done or would be more suitable in future developments of Topo-seq with larger DNA libraries.

Following the revision, we expect the reviewers will find that our study clearly demonstrates that nucleosomes have an intrinsic capacity to restrain DNA topology, which is imprinted by their native genomic loci; and that the Topo-seq approach presented in our study opens the possibility of uncovering the interplay of chromatin and DNA topology genome-wide.

Reviewer #1 (Remarks to the Author):

Segura et al. report that DNA associated with nucleosomes restrain different ΔLk (linking number difference) depending on their genomic localization. This work follows up their previous report of a ΔLk of -1.26 for the bulk of nucleosomes in *S. cerevisiae*. Briefly, the authors cloned mononucleosomal DNA in a plasmid (Figure 1) and measure the ΔLk after its introduction in *S. cerevisiae* either in a few individual molecules (Figure 2) or in two subsets of the whole library fractionated by the differential mobility of the topoisomers (Figure 3). The work is potentially interesting to a wide readership but, in my opinion, the results are preliminary because several additional controls would be necessary to support the conclusion that nucleosomes confer a different ΔLk depending on their genomic localization (Figure 4).

1. To isolate mononucleosomal DNA, the authors digest chromatin with micrococcal nuclease (MNase) and recover the band of ~ 150 bp from a gel (Figure 1a). As a quality control of nucleosomal DNA, they estimate that $>91\%$ of the recovered fragments overlap at least 50 bp to previously reported nucleosomes. I think that a more rigorous control would be necessary, given that 50 bp is a third of nucleosomal DNA and many non-nucleosomal fragments 150 bp long would also overlap > 50 bp to nucleosomal DNA. The recovered MNase fragments should be sequenced and aligned to the *S. cerevisiae* reference genome to generate a nucleosome occupancy map. Its comparison to other good quality maps in the literature would guarantee the nucleosomal origin of the recovered fragments. This control is important to ensure a homogeneous coverage of the genome. It is very common that AT- or GC-rich fragments are over or underrepresented in the final dataset depending on a) the degree of digestion with MNase, b) biased enrichment caused by some kits during the preparation of the sequencing libraries and c) differential amplification during the PCR step.

RESPONSE

We thank the reviewer for pointing out issues frequently arising in nucleosome mapping studies. One can never discard that a small fraction of the ≈ 150 bp fragments typically obtained upon digesting chromatin with MNase could be non-nucleosomal. We already tackled this matter in our previous report of a ΔLk^{nuc} of -1.26 for the bulk of nucleosomes in *S. cerevisiae* (Segura et al. Nature com 2018). In this paper, we showed that

inserting the nucleosome library in the circular minichromosome produces the expected MNase footprint of an additional nucleosome concomitant to an average change of $-1.26 \Delta Lk$ units. In our present study, we observed again this gain of $-1.26 \Delta Lk$ units and found little dispersion of the individual ΔLk^{nuc} values around this mean value. All these observations denote that most library fragments must assemble a nucleosome. **We remarked this notion in the results section related to Fig. 2a-b.**

After sequencing and obtaining the genomic coordinates of the nucleosomal DNA library fragments, we used the catalogue of Jiang and Pugh (2009) to assign nucleosome IDs to our library (Supplementary Data 1). This catalogue compiled six sets of nucleosome maps generated from different laboratories using different technologies. Considering the experimental variability among labs and the positional fuzziness of many nucleosomes, the correspondence of our library with these previously referenced coordinates was significant. We illustrated this correspondence in the revision by plotting the degree of overlapping of our fragments with this catalogue (Supplementary Fig. 2a); and by comparing the overlapping of our library and that of a random set of coordinates (Supplementary Fig. 2b). Lastly, we also included in the revision a plot showing that the library degree of overlapping with the catalogue has no correlation with the ΔLk^{nuc} values, which further substantiates that most library fragments assembled a nucleosome (Supplementary Fig. 6).

Supplementary Fig. 2. Correspondence of the nucleosome library coordinates with previously referenced nucleosomes. **a**, Degree of overlap (number of bp) of the nucleosomal DNA library coordinates with those of the catalogue of Jiang and Pugh (2009), which compiled six sets of nucleosome maps generated from different laboratories using different technologies. Nearly all (>91%) of the library overlapped (>50 bp) with the genomic coordinates of previously referenced nucleosomes. **b**, Comparison of the overlapping of the referenced nucleosomes with the library and a random set of coordinates. The overlap ratio of 0 means absolutely no overlap and 1 means full overlap. **c**, GC-content (%) of the nucleosomal DNA library and that of the full catalogue of nucleosomes.

Supplementary Fig. 6. Correlation of ΔLk^{nuc} and nucleosome overlapping with the reference catalogue. The plot shows ΔLk^{nuc} values obtained via Topo-seq against the extent of overlap (bp) of the nucleosomal DNA fragments with previously referenced nucleosomes in Jiang and Pugh (2009). The trend line of the average ΔLk^{nuc} value is depicted (red).

Regarding possible deviations caused by MNase digestion or bp composition, we clarified in the results section that to construct the library "... we digested budding yeast chromatin with Micrococcal nuclease and purified a pool of mono-nucleosomal DNA fragments (≈ 150 bp) that combined increasing digestion rates...". Therefore, we minimized a plausible enrichment of more or less accessible nucleosomes. Accordingly, the relative abundance of nucleosome classes described in Fig. 1b-d and Fig. 5a-c denotes that the digestion generated a representative library. We also remarked in the results section related to Fig. 3a the following: "... note that since ΔLk^{nuc} calculations derive from the partition ratio of each sequence in these gel sections, potential biases due to the length and bp composition of nucleosomal DNAs would similarly occur in both sides and thus have little effect on the partition ratios...". Concerning possible biases produced during DNA amplification and sequencing, we tested whether AT- or GC-rich fragments were over or underrepresented in the final dataset. We found that the GC% of the nucleosomal DNA library (41.5%) was slightly higher than that of the reference catalogue (40.8%) (Supplementary Fig. 2c). This difference could be attributed to the overrepresentation of rDNA nucleosomal sequences (45% GC) in our library, as we explain below (point 5).

2. To calculate the ΔLk of nucleosomes (ΔLk_{nuc}), the authors subtract the Lk of the empty plasmid (Lk_{Chr}) (1341 bp) from the Lk of the plasmid harbouring nucleosomal inserts ($Lk_{Chr+nuc}$) (1341 + ~ 150) (Supplementary Fig. 2). This assumes that the difference in size between the plasmids will not affect their topology and, therefore, the ΔLk_{nuc} will be a direct consequence of the topological properties of the nucleosomal insert. A possible control to clarify this point would be to generate in the same plasmid a library of random fragments ~ 150 bp derived from, for example, naked *S. cerevisiae* DNA partially digested with MNase or from DNA of prokaryotic origin having the same G+C content as *S. cerevisiae*. If these control libraries have the same ΔLk as the empty plasmid, it could be concluded that $\Delta Lk_{Chr+nuc}$ is really due to the nucleosomal inserts. However, if ΔLk of the control libraries is closer to $\Delta Lk_{Chr+nuc}$, it would be necessary to revise the conclusion that it was due exclusively to the properties of nucleosomal DNA.

To calculate ΔLk^{nuc} , we do not directly subtract the Lk of the empty YCp1.3 minichromosome (Lk^{Chr}) from that of the minichromosome harbouring nucleosomal DNA inserts ($Lk^{Chr+nuc}$). In such a case, Lk differences would be of about 14 units (≈ 150 bp/10,5). ΔLk^{nuc} is calculated as the difference between ΔLk^{Chr} and $\Delta Lk^{Chr+nuc}$, each of which is first calculated by subtracting the Lk of the relaxed naked DNA (Lk^0) from that of the corresponding chromatinized DNA (Lk^{Chr}). We explained this notion in more detail in Supplementary Fig. 3.

We understand that the reviewer is also concerned about the possibility that the chromatin structure of the minichromosome and, therefore, its global DNA topology could be altered upon inserting an additional ≈ 150 bp. We already considered this possibility in our previous studies (Diaz-Ingelmo et al. 2015; Segura et al. 2018), in which we designed the insertion point in between two well-structured and positioned elements: the centromere (CEN) and the nucleosome V that flanks ARS1. In this way, we minimized plausible interferences of the nucleosome library with the functional units of the minichromosome (as explained in Supplementary Fig. 1). Suitably, we showed that the nucleosome organization (MNase digestion map) of the YCp1.3 minichromosome backbone remains unchanged after inserting a nucleosome library. Moreover, our previous and present studies denote that ΔLk restrained by the point centromere (+0.6) and the segment I to V (-6.4) of YCp1.3 does not change after inserting either the library or individual nucleosomes. We included these Lk values to illustrate this last point in Supplementary Fig. 3.

Supplementary Fig. 3. Experimental layout to calculate the ΔLk restrained by nucleosomes *in vivo*. In a circular minichromosome (i), the DNA linking number difference constrained by its chromatin elements (ΔLk^{Chr}) is calculated by comparing, via gel electrophoresis (ii), the Gaussian distribution of Lk topoisomers of the DNA relaxed *in vitro* (lane 1) with that of the minichromosome DNA fixed *in vivo* (lane 2). The gel position of the Lk mean of the relaxed DNA (Lk^0) and that of the minichromosome DNA (Lk^{Chr}) is determined by plotting the intensities of their corresponding Lk topoisomers along a scale of ΔLk units (iii). ΔLk^{Chr} is the distance in Lk units between the two means ($Lk^0 - Lk^{Chr} = \Delta Lk^{Chr}$). In the YCp1.3 minichromosome, the ΔLk^{Chr} value of -5.81 results from the ΔLk restrained by the point centromere (+0.6) and the ΔLk restrained by the segment I to V (-6.4). Upon adding a new nucleosome (≈ 150 bp) (iv), the absolute Lk of the DNA circle increases by about 14 units (≈ 150 bp/10.5 bp per helical turn). The gel position of the Lk mean (Lk^0) of this larger DNA and the corresponding minichromosome DNA ($Lk^{Chr+nuc}$) are again determined to obtain $\Delta Lk^{Chr+nuc}$. The resulting difference between ΔLk^{Chr} and $\Delta Lk^{Chr+nuc}$ equals ΔLk^{nuc} , the ΔLk restrained by the added nucleosome (v). ΔLk^{nuc} is about -1.26 for most nucleosomes, as previously reported by Segura *et al* (2018).

Lastly, we considered doing the control experiment proposed by the reviewer, but we realized that this construction is not trivial. Moreover, this experiment would only be informative as long as we could create a library of random ~ 150 bp fragments that absolutely exclude nucleosome assembly and any other DNA-protein interaction. However, former chromatin studies demonstrated that partial or near-complete nucleosome assembly can occur in most DNAs, including those of prokaryotic origin. Nevertheless, since there is no evidence of structural alterations in the hosting minichromosome after inserting the library, we believe that the assumption that ΔLk^{nuc} values reflect the topological properties of the inserted nucleosomes is the most simple and reasonable; and also the only able to explain the correlation of ΔLk^{nuc} with the genomic origin of the inserted nucleosomes.

3. The fragments analyzed are 144 ± 21 bp long (page 4). This means that the same 147 bp of nucleosomal DNA could have heterogeneous flanking sequences in the population of fragments due to variability of MNase digestion in different cells. It would be important to assess the effect of this heterogeneity in the ΔLk . A possible experiment to test it directly would be to generate several nucleosomal sequences 147 bp long from specific well positioned nucleosomes in the genome with flanks of different sizes. They should be cloned in the plasmid to test their Lk individually as in Figure 2 to measure directly the effect of the flanks on the same nucleosomal sequences. The result could be used to confirm the more indirect estimate of 0.007 Lk units for each bp difference in length obtained by subtracting the average ΔLk of all sequences of the same length from the ΔLk score of each individual sequence (page 6).

Certainly, we did not directly test whether linker DNA length could affect Lk^{nuc} . However, in our previous study, we had designed the insertion point of the nucleosome library to provide a spacing of about 65 bp on the CEN side and 38 bp on the Nuc-V-ARS side of YCp1.3, as illustrated in Supplementary Fig. 1. Therefore, nucleosome assembly should not have restrictions due to an insufficient linker length.

To further assess the effect of linker length on Lk^{nuc} , we have conducted the experiment proposed by the reviewer and included it in the revision (Supplementary Fig. 7). Namely, we constructed minichromosomes with a nucleosomal DNA insert of 147 bp plus 0, 10 or 20 bp of extra linker DNA at each flank. We calculated the Lk^{nuc} produced by these 3 constructs following the procedure described in Supplementary Fig 3. We found that Lk^{nuc} was -1.26 for the 147 bp insert, -1.27 for the 167 bp insert, and -1.28 for the 187 bp insert. Therefore, Lk^{nuc} was barely affected by the linker length. **We comment this new observation in the results section related to Fig 3h.**

The above experiment also exposed that increasing the linker DNA length reduced the gel velocity of the DNA constructs. As proposed by the reviewer, we measured these differences and found that the Lk mean of the 167 and 187 bp constructs was retarded about a distance equivalent to +0.15 and +0.30 Lk units with respect to that of the 147 bp construct (Supplementary Fig. 7c). These values are in good agreement with and thus validated the 0.007 Lk unit/bp correction used in our study. Without this length correction, the Topo-seq data would have produced a Lk^{nuc} of -1.26 for the 147 bp, -1.13 for the 167 bp, and -1.00 for the 187 bp.

Supplementary Fig. 7. Effect of the linker DNA length on Lk^{nuc} . **a**, Lk distributions of YCp1.3 hosting a nucleosomal DNA insert of 147 bp plus 0, 10 or 20 bp of extra linker DNA at each side (147, 167, 187 bp inserts). Nucleosome coordinates are described in Supplementary Table 1. Lanes 1-3 show the relaxed DNA Lk distributions of these constructs, and lanes 4-6 show those of the corresponding minichromosomes. **b**, Topoisomer intensities, $\Delta Lk^{Chr+nuc}$ values and standard deviation (SD) of the minichromosome Lk distributions. ΔLk^{nuc} produced by each insert (147, 167, 187 bp) was obtained as the difference between the $\Delta Lk^{Chr+nuc}$ values and -5.81, which is the ΔLk^{Chr} of the empty YCp1.3 minichromosome. Lk^{nuc} was -1.26 for the 147bp insert, -1.27 for the 167bp insert, and -1.28 for the 187bp insert. **c**, Gel densitometry of lanes 4-6 and position of the Lk means (red lines). The Lk mean of the 167 and 187 bp constructs (lanes 5 and 6) was retarded about a distance equivalent to +0.15 and +0.30 Lk units relative to that of the 147 bp construct (lane 4). These values are in good agreement with the 0.007 Lk unit/bp correction used in our study.

4. To fractionate the distribution of topoisomers at higher resolution, the gel in Figure 3a could have been sliced in several fractions, not only in two. Increasing the resolution of the fractionation is very important because the relative representation of each fragment on each gel fraction is used to assign a specific Lk to each fragment (Suppl Table 3).

Undoubtedly, increasing the resolution of the fractionation, for instance by cutting the gel lane into numerous slices of 1 mm, would provide more accurate results and also allow analysis of DNA rings of multiple lengths (1 to 15 Kb) in a single gel lane. We are exploring how to perform this fractionation for future developments of Topo-seq, as well as the possibility of running two-dimensional gels. **We have highlighted these possibilities in the revised discussion (second paragraph).** However, in this initial study, we aimed to present the Topo-seq

approach as simply as possible. Therefore, since the length differences produced by the library were small compared to the size of the minichromosomes (1.5 kb), we cut the overlapping Lk distributions into just two halves. We then corrected the gel migration differences among Lk distributions, which were always smaller than the gel distance between individual Lk topoisomers (see also our response to reviewer#3 point 1).

5. Differences in the mean ΔLk of nucleosomes on different genome locations (Figure 4) are very subtle and, given the concerns raised in the previous points, they should be experimentally confirmed. For example, some nucleosomal sequences from these different regions could be cloned in the same 1341 bp plasmid and their ΔLk should be determined individually to confirm the location dependent differences described in Figure 4. Although the genome coverage of nucleosomal fragments seems to be uniform across chromosomes (Figure 1b), a possible concern in Figure 4 is that the proportion ribosomal DNA fragments (2043 out of 8998 = 22.7%) is more than 2-fold the approximately 10% of the genome represented by the rDNA genes. Also, the ΔLk of +1 nucleosomes is based on 403 nucleosomes only. Considering a conservative number of 5000 protein coding genes in *S. cerevisiae*, the analysis represent approximately 8% of all +1 nucleosomes.

We agree that to validate the Topo-seq results and conclusions of our study, they had to be corroborated by measuring the ΔLk^{nuc} of individual nucleosomes directly. We conducted thus the experiment proposed by the reviewer as follows: We chose 18 nucleosomes of the library (3 gene body, 3 intergene, 3 rDNA, 3 telomeric, 3 nuc+2, 3 terminal), whose ΔLk^{nuc} values determined via Topo-seq were representative of their corresponding regions. We amplified and inserted these sequences (147 bp in length) in YCp1.3, transformed yeast cells and examined the Lk distributions of the minichromosomes to determine the ΔLk^{nuc} as described in Fig. 2. These individually measured ΔLk^{nuc} values presented a good correlation ($R^2 \approx 0.9$) with those calculated via Topo-seq. These new results confirm the location-dependent topology of nucleosomes and the feasibility of Topo-seq to conduct these analyses. We included these experiments in the revision as a new section of the results, Fig. 5d and Supplementary Fig. 10.

Fig. 5 - ΔLk correlation with the native genomic allocation of nucleosomes. **a**, Boxplot of ΔLk restrained by gene body, intergenic and rDNA nucleosomes. p -values determined by ANOVA with Tukey's test. **b**, Boxplot of ΔLk restrained nucleosomes at positions -1,+1,+2, +3 to +5 relative to the TSS and the TTS (Term). p -values determined by ANOVA with Tukey's test. **c**, Boxplot of ΔLk restrained by non-telomeric and telomeric nucleosomes. p -value determined by unpaired two-tailed t -test. In **a-c**, the bottom rows indicate the number (n), average sequence length (bp) and mean ΔLk of

nucleosomes in each class. The ΔLk^{nuc} and genomic allocation of individual nucleosomes are described in Supplementary Data 3. **d**, Correlation of the ΔLk^{nuc} values obtained via Topo-seq with those calculated via the analyses of individual Lk distributions (see Supplementary Fig 10). The scatter plot shows the Lk^{nuc} of 18 representative nucleosomes (3 gene body, 3 intergenic, 3 rDNA, 3 telomeric, 3 nuc+2, 3 terminal) obtained by both procedures. Nucleosome coordinates and ΔLk^{nuc} values are specified in Supplementary Table 1.

Supplementary Fig. 10. Validation of the ΔLk^{nuc} dependence on the genomic origin of nucleosomes. **a**, Gel electrophoresis of Lk distributions of the YCp1.3 minichromosome hosting 18 nucleosomal DNA sequences of 147 bp, whose ΔLk^{nuc} determined via Topo-seq was representative of the mean ΔLk^{nuc} value of their corresponding allocations (gene body, intergenic, rDNA, telomeric, nuc +2, terminal nucleosome). The chromosomal coordinates and PCR primers for cloning these 18 nucleosomal DNA sequences into YCp1.3 are described in Supplementary Table 1. Electrophoresis conditions are described in Methods. Arrowheads indicate Lk topoisomers. **b**, Gel densitometry and relative topoisomer intensities of the previous Lk distributions (lanes 1-18). The ΔLk^{nuc} restrained by each of the 18 nucleosomes was calculated relative to the reference nucleosome in line 1 ($\Delta Lk^{nuc} = -1.26$), which is the same nucleosome used to test the effect of the linker DNA length on Lk^{nuc} (Supplementary Fig. 7).

Our library represents approximately 8% of the yeast nucleosomes (about 60.000) because we could only collect ~10.000 yeast transformants to perform this study. However, as the reviewer acknowledges, the coverage of our library seems to be uniform across chromosomes and genomic locations except for the rDNA nucleosomes, which are overrepresented by over 2-fold. This enrichment occurred because, following Topo-seq, we excluded the nucleosome DNA sequences with few reads, and this filter did not affect the rDNA nucleosomes, which are overrepresented by nearly two orders of magnitude relative to others. However, note that this 2-fold enrichment of rDNA nucleosomes does not affect the calculation of Lk^{nuc} , because the partition ratio of each sequence in the gel sections will be similar regardless of whether a sequence is over or underrepresented in the loaded pool. We commented on this notion in the revised discussion (third paragraph):... “Since the calculation of ΔLk values derives from the distribution of each sequence in different gel sections, these ratios can be calculated regardless of the abundance of each sequence in the library pool”...

6. As mentioned in point 1, small differences in the level of MNase digestion can cause significant variations on the quality and relative enrichment of nucleosomal DNA. For this reason, it should be tested how robust the differences reported in Figure 4 are to differences in the level of digestion with MNase. The Methods section indicates that mononucleosomal DNA fragments were eluted from the gel and pooled. Does pooling refer to the three lines of the gel in figure 1a? Would comparable results be obtained in duplicate experiments using mononucleosomal fragments from an independent experiment? The same reasoning applies to the cutting of the gel slab at the level of ΔLk of exactly -1.26 in the global distribution of topoisomers (Figure 3a). It seems that no replicate of this experiment was done. In other words, how robust are the results obtained to experimental variation?

We believe that in our study any potential bias caused by differences in the level of MNase digestion can be excluded for the following reasons: First, we pooled mononucleosomal DNA fragments obtained with different degrees of MNase digestion, as those in the three gel lines in Fig. 1a. Therefore, the pool of 150 bp fragments included those nucleosomes quickly accessible and those more resistant to MNase digestion. We clarify this issue in the revised text. Second, the representative coverage of the yeast nucleosomes described in Fig. 1b-d and Fig. 5a-c denotes that chromatin digestion did not create a bias in this regard. Third, the calculation of ΔLk^{nuc} depends only on the partition ratio of each DNA sequence at each side of the gel cut and, so, with an equal degree of digestion. Lastly, note that to calculate ΔLk^{nuc} via Topo-seq we normalized Z-scores of the nucleosomal DNAs only with those of the same length. Therefore, ΔLk^{nuc} differences cannot be consequent of comparing more- and less-digested types of nucleosome. In this regard, our finding that nucleosomes from telomeric regions have less negative ΔLk^{nuc} values could be a priori attributed to their shorter DNA sequences. However, the singular topology of telomeric nucleosomes remains significant even when compared to short nucleosomal DNAs of non-telomeric regions. **We commented on this in the results related to Fig. 5c and Supplementary Fig. 9.**

Supplementary Fig. 9. Lk^{nuc} of telomeric and short no-telomeric nucleosomes. Boxplots of ΔLk^{nuc} values restrained by telomeric nucleosomes and non-telomeric nucleosomal DNAs with fragment lengths 135 to 140 bp. p -value determined by unpaired two-tailed t -test.

As the reviewer pointed out, we could produce an experimental error if the gel was not cut exactly at the level of ΔLk^{nuc} -1.26. However, note that such an error would not alter the classification of the nucleosomes by their capacity to restrain ΔLk and, therefore, that would not change the conclusions of the study. Nevertheless, after comparing the ΔLk^{nuc} values calculated via Topo-seq with those measured in individual nucleosomes (Fig. 5d), we can conclude that the cut was indeed at about the level of ΔLk^{nuc} -1.26. Regarding the robustness of our results under different experimental conditions, we emphasized in the discussion (last paragraph) that all the nucleosomes of the library are compared in the same environment (the minichromosome), not in their native loci, where experimental conditions could differentially affect their topology via genome activities and chromatin dynamics. Therefore, the ΔLk^{nuc} values that we measured reflected an intrinsic property of the nucleosomal DNA sequences.

Reviewer #2 (Remarks to the Author):

In this study, the authors introduce Topo-seq, a method to calculate linking number difference (dLk) introduced by ~150 bp fragments (genome-wide) into a reference plasmid. dLk is calculated usually using gel electrophoresis, but doing so for 1000s of possible sequences is almost impossible. Here, the authors elegantly demonstrate a sequencing based method that allows electrophoretic separation of a library of 1000s of plasmids with inserts corresponding to ~150 bp of DNA protected by nucleosomes in the yeast genome. They find +2 nucleosomes of genes and telomeric DNA to have significantly altered dLk compared to the rest of the genome, hinting at the intrinsic DNA sequences driving distinct chromatin structures in these regions. I have the following comments:

1. The authors do not discuss replicates. How reproducible is the dLk values from a genome-wide assay?

RESPONSE

In our previous study (Segura et al. 2018), we inserted a library of about 1000 nucleosomes into the YCp1.3 minichromosome and found that the mean ΔLk restrained by the nucleosomes was about -1.26. In our present study, we constructed a larger nucleosome DNA library and found again a mean Lk^{nuc} of -1.26. These average ΔLk^{nuc} values were thus reproducible. We conducted the Topo-seq procedure with the second library to obtain the classification of nucleosomes in terms of their intrinsic capacity to restrain ΔLk in a common environment (the YCp1.3 minichromosome). Namely, as we explain in response to reviewer#1, the classification of the nucleosomes via Topo-seq should persist regardless of potential variability in the degree of chromatin digestion, experimental conditions, or experimental errors in cutting the gel. Another question is how accurate are the ΔLk^{nuc} values within the classification. In this respect, the validation of Lk^{nuc} values conducted with 18 individual nucleosomes (Figure 5, Supplementary Fig. 10, Supplementary Table 1) indicates that the average ΔLk^{nuc} values that characterize different nucleosome classes are reproducible and quite accurate.

2. Related to point 1, the authors should compare the dLk values obtained from their assay to the ones determined by gel electrophoresis for a few nucleosomes (either from previous studies or select a few from this study). This will again validate the genome-wide dLk calculations. This is essential especially since the authors introduce a statistical approach with several steps to calculate the final dLk from their sequencing datasets and it has to be shown that their calculations yield accurate dLk values.

RESPONSE

We agree on the importance of conducting these validation experiments, also proposed by the two other reviewers. As indicated above, we included these experiments in Fig.5, Supplementary Fig. 10 and Supplementary Table 1 of the revision. See our response to reviewer #1 for details. The consistency of the individual ΔLk^{nuc} values of 18 nucleosomes with those obtained via Topo-seq supports their reproducibility and validates the statistical approach to calculate them.

3. Although the authors mention dLk of "nucleosomes" throughout the text, we do not know if the sequences form nucleosomes on the plasmids. Similar to point 2, it would good to validate that insertion of a random 150 bp sequence that protected a nucleosome in the genome would result in the plasmid actually having an extra nucleosome.

RESPONSE

There are several indications that the library of DNA fragments forms nucleosomes (see our responses to points 1-3 of reviewer#1). Namely, in our previous study (Segura et al. 2018), we designed an insertion point of the nucleosome library in YCp1.3 to provide a spacing of about 65 bp on the CEN side and 38 bp on the NucV-ARS side (Supplementary Fig. 1). Therefore, the DNA fragments obtained by MNase digestion of chromatin should be able to assemble a nucleosome without restrictions due to an insufficient linker length. Accordingly,

in this previous study, we also showed that inserting the library produces the expected MNase footprint of an extra nucleosome at the insertion point and generates a concomitant gain of about $-1.26 \Delta Lk$ units in the minichromosome. In our present results with Topo-seq, the dispersion of ΔLk^{nuc} around the mean value (-1.26) further denotes that most library fragments must assemble a nucleosome. We can not envision other mechanisms able to produce such a general gain of $-1.26 \Delta Lk$ units.

4. The genome-wide analysis is very superficial. Some analyses that come to mind that haven't been performed is, what is the correlation between transcription and dLk at +1, +2, and gene body? How does dLk change at transcription factor binding sites? How about centromeres? Is the in vivo linker length reflected by intrinsic dLk values? The authors need to think about how to make their genome-wide analysis more substantial.

RESPONSE

Undoubtedly, the analyses derived from Topo-seq or any other genomic data can potentially be much deeper. However, this is the first study conducted using this new approach and the library size we obtained was quite limiting to produce more statistically significant correlations. Therefore, we focused the discussion and conclusions on the statistically robust findings. So far, we can safely conclude that nucleosomes have an intrinsic capacity to restrain DNA supercoils, which is imprinted by their native genomic loci.

Regarding further analyses like those proposed by the reviewer, we looked for correlations of Lk^{nuc} with gene transcription but could not find clear significant differences to include in the present report. For instance, we observed that ΔLk^{nuc} values tend to be less negative in TATA genes than in the TATA-less ones. We observed also that ΔLk^{nuc} tends to be more negative in late-replicated regions. We expect that, in future studies with larger libraries, we should be able to look deeper into these aspects and also on the effect of TFBS on nucleosomal DNA topology. **We comment on that in the revised discussion (last paragraph)**. Regarding centromeres, since YCp1.3 is already a centromeric minichromosome, insertion of a second centromere would be excluded. Accordingly, we did not find centromeric sequences in the library. However, we already described the DNA topology of yeast point centromeres in a previous study (Diaz-Ingelmo et al 2015). Regarding the effect of linker length, see **Supplementary Fig. 7** and our response to point 3 of reviewer#1.

Reviewer #3 (Remarks to the Author):

The authors present original research, which demonstrates that nucleosomal DNA topology is imprinted into DNA sequence. To achieve this, the authors placed a library of nucleosome-binding DNA fragments, obtained from MNase digestion, in the YCp1.3 yeast circular minichromosome. The library was amplified in yeast cells and run on gel electrophoresis. Due to varying size of library fragments, the authors obtained smeared distribution instead of clear separate bands expected for DNA topoisomers. The smearing prohibited direct quantification of topoisomers. To solve this problem the authors cleaved gel in two halves with an assumption that the mean of the smeared distribution corresponds to the average $\Delta Lk = -1.26$ per nucleosome (as measured in the previous publication of the same group), where Lk is a linking number. By sequencing minichromosomes extracted from both sides of the cleaved gel and counting the relative abundance of each library fragments on both sides of the gel, the authors built Z-score function to quantify ΔLk of individual library fragments. The authors performed statistical analysis and showed that specific values of ΔLk are enriched in different genomic regions (gene promoters, ± 1 nucleosome around TSS, gene bodies, etc.), which made authors to conclude that topology of the DNA wrapped around nucleosome has specific features that are encoded solely by DNA sequence and persists when relocated into different genomic context (i.e. into circular minichromosome).

While this is definitely an interesting study, it has a few problematic places in my opinion. It also lacks rigorous description of the method author used. This should be addressed before publishing.

1. I worry about potential bias from the fragment library size variability. The authors acknowledge that topoisomer's ladder is not aligned across multiple library fragments due to different fragment size. It could be a problem in my opinion if the shift in migration distance due to different fragment size is more than distance between two topoisomers. In this case you can get all topoisomers of a specific fragment on one half of the gel just because it migrates much slower or much faster due to the fragment size difference. At Fig. 2b the authors show how topoisomer peaks shift due to difference in the fragment size. They compare fragments in the range of 144-150 bp and see significant shift, which is however within acceptable range. However, the size of their library is quite broad 144+/-21 bp. Thus, it would be great to see how fragments randomly chosen from the library with a broad range of size distribution behave on gel. It would also be beneficial to see how the fragments picked from extremes of Z-scores value behave on gel. To reduce fragment library size bias the authors can try to restrict size of the fragments to the narrow window and sequence the library deeper to get enough statistics.

RESPONSE

The fragment size variability of the nucleosomal DNA library was a central matter that we carefully considered to calculate the ΔLk^{nuc} values. In this respect, we minimized any potential bias due to the fragment size variability based on the two reasons that produced the misalignment of the Lk distributions:

The first is that the helical phasing of Lk topoisomers of a DNA circle changes for each additional bp in length until this bp number is a multiple of the DNA helical repeat (≈ 10.5 bp). Therefore, two DNAs differing in <10 bp will present misaligned Lk ladders although they can have equal Lk mean (Supplementary Fig. 4). We solved this problem by normalizing all the ΔLk scores with respect to inserts of equal length (Fig. 3e). Therefore, we excluded any bias in this regard.

The second reason is that DNA size differences affect the overall velocity of the Lk distributions. We calculated that each bp produces a shift in the velocity equivalent to 0.007 Lk units (Supplementary Fig. 5). In this respect, we appreciate the control experiment proposed by the reviewer, which is equivalent to that suggested by reviewer#1 (point 3). The results confirmed the accuracy of the 0.007 Lk units/bp correction to compare ΔLk distributions of different DNA length (Supplementary Fig. 7). This experiment also illustrated that the shift in migration due to library size variability (144 ± 21 bp) is always smaller than the distance between two topoisomers. **We remarked on this in the revised discussion (second paragraph)**. Namely, only a length difference of more than 140 bp would surpass the distance between two topoisomers. Therefore, we also excluded any significant bias in this regard.

2. At Fig. 3 c-f, the authors partially try to assess the bias due to fragment library size variability. They write "Although section B slightly enriched in short nucleosome DNA sequences, the partition probabilities within each group denoted main differences in nucleosome DNA topology". It is hard to agree with term 'slightly' here. As an example, at Fig. 3c the difference in A/B ration varies by factor of 10 depending on fragment size. However, due to log scale this is hard to see. Also, to support statement that partition probabilities denote the main difference in DNA topology, the authors should pick up certain fragment length, randomly select representative number of fragments, do gels and compare to their partition function scores.

RESPONSE

We agree the term "slightly" was misleading in describing Fig. 3c. Thus, we modified these sentences as follows: **"Section B was enriched in short nucleosomal DNA sequences (<140 bp) since the Lk distributions of the corresponding DNA rings migrated faster during electrophoresis. However, the partition ratios A/B varied up to a factor of 10 within each length group, which denoted a diversity in nucleosome DNA topology"**.

To confirm that partition probabilities mainly denote differences in DNA topology, we conducted the validation experiment proposed by the reviewer (also proposed by reviewers#1 and #2). We examined individually the ΔLk^{nuc} of 18 nucleosome DNAs of equal length that were representative of the distinct genomic loci described in the study. The ΔLk^{nuc} of these nucleosomes were consistent with those obtained via Topo-seq, thus

validating our approach to calculate ΔLk^{nuc} from partition probabilities. These confirmatory results are in Fig. 5, Supplementary Fig. 10 and Supplementary Table 1 of the revision. See also our response to reviewer#1 for details.

3. Normalization of Z-score by 0.85, which as authors wrote is the standard deviation of Lk distributions of the DNAs hosting nucleosome library, can be problematic since it doesn't take into account shift due to fragment size variability.

RESPONSE

The effect of fragment size variability on the standard deviation of Lk distributions of the minichromosome DNAs hosting the library is very small. Note that the SD of an Lk distribution depends on the total length of the DNA circles, which is about 1.5 kb in our study (1341+144 ±21 bp). Therefore, SD values were very similar (about 0.85) for all minichromosomes regardless of fragment size variability. This similarity was corroborated when testing the effect of the linker DNA length on Lk^{nuc} (Supplementary Fig. 7). Note also that small changes of SD due to fragment size variability would have a negligible effect on the Lk^{nuc} classification of the nucleosomes since we normalized the ΔLk scores relative to inserts of equal length.

4. At Fig. 4h the authors try to interpret observed ΔLk values in terms of the geometry of DNA wrapped around nucleosomes. While this is potentially possible interpretation, I'm wondering if the authors tried to consider other interpretations, which could be partial nucleosome occupancy of library fragments, or activity of remodeling complexes, which can change frequency of the observed topoisomeric states without changing DNA topology?

RESPONSE

We thank the reviewer for raising this issue. In Fig. 6 (old Fig. 4h), we interpreted the ΔLk^{nuc} values as different extends of DNA wrapping. However, we agree that other possibilities, as those proposed by the reviewer, can not be discarded. These alternative mechanisms could affect specific nucleosomes but will still depend on the DNA sequence since we interrogated the nucleosome DNA topology outside their native context. **We commented on this matter in the revised discussion (last paragraph).**

Minor comments:

1. The authors frequently use term 'nucleosome library' across the text, which is a bit misleading. I believe the authors should stick to 'nucleosome DNA library'.

Done

2. Fig 1 legend: "In c and d, the nucleosome library (n=4276, pink) is compared to the full catalogue of the yeast nucleosomes." However, there are not pink color in those panels.

Done

3. Page 6: "as in the cannon nucleosome". Should be canonic.

Done

Reviewers' Comments:

Reviewer #1:

Remarks to the Author:

Point 1.

I suggested that sequenced mononucleosomal DNA should be mapped onto the reference genome to generate a genome-wide nucleosomal occupancy map to be compared to previously reported maps. Alternatively, mapping could have been done with nucleosomal DNA represented in the library to assess its quality relative to previous studies. This is a very straightforward control that does not exclude the control that the authors did by overlapping the sequences of the library to a catalog of nucleosome positions previously reported (new Suppl Fig 2a). An important negative control missing in that figure is the overlap to the same intervals of the same number of fragments 150 bp long selected at random bioinformatically in the *S. cerevisiae* genome. Results should be shown in histogram bars next to those of the library in the same figure for direct comparison of the enrichment of the library in nucleosomal DNA. In Suppl Fig 2b, the authors compare the overlap with random coordinates in the genome but, since most of the genome is complexed with nucleosomes, it is not clear how this comparison informs of the enrichment of the library in nucleosomal DNA. Again, as a negative control, the overlap ratio of the collection of 150 bp random fragments with the same random coordinates should be shown next to the random coordinates boxplot.

Point 2.

Related to the previous point, I suggested negative controls to make sure that the observed differences in ΔLk between nucleosomal DNA from different genomic locations (which is the main point of the work) were specific of nucleosomal DNA. In the absence of this control, it cannot be guaranteed that the ΔLk properties they attribute to nucleosomal DNA are not also detectable in non nucleosomal DNA. The fact that the mean value of ΔLk of -1.26 is the same in DNA fragments that overlap >90% as in those overlapping only 33% (50 bp) to the reference catalogue of nucleosomes (new Suppl Fig 6) leaves open the possibility that the -1.26 value could not be an exclusive property of nucleosomal DNA. Therefore, it is essential to make a library of non nucleosomal DNA (\sim 150 bp fragments of naked DNA, even of prokaryotic origin with similar base composition to *S. cerevisiae*, partially digested with MNase) to compare the mean distribution of ΔLk with that shown in the new Suppl Fig 6. The authors claim that such library would not exclude that some nuclear proteins or nucleosomes could bind to them. This is true, but it would be unlikely that the mean ΔLk of an entire non nucleosomal library, or of a representative sample of individual fragments, would be close to -1.26 if that value is specific of nucleosomal DNA.

In their reply, the authors explain that prior to estimating the differences between ΔLk_{Chr} and $\Delta Lk_{Chr+nuc}$ they first subtract the Lk of the relaxed minichromosome from that of the chromatinized counterpart (Suppl Fig 3). That approach was used in a previous article (Segura et al. 2018) but not in this work where, unless I am missing something, the topoisomerase relaxed minichromosome library was not used as a reference. Instead, Lk was estimated from the ratio between the representation of the individual DNA fragments in two gel fractions (Figure 3).

Point 3.

The authors have shown that variations in the size of linker DNA do not affect ΔLk (Suppl Fig 7). Unlike in the analysis of the entire library (Figure 3), in this case, they estimate ΔLk using the ratio between the chromatinized and relaxed versions of the minichromosome.

Point 4.

Since the authors show that the length of the DNA fragments inserted in the minichromosome does not alter the ΔLk after applying the 0.007 Lk unit/bp correction (Suppl Fig 7), how can it be explained that overlapping fragments that differ in one or a few bp generate a large difference in their ΔLk . Just to mention a couple of examples on page 1 (out of 86 pages) in Table 3, the two first fragments starting in position 26200 that differ in one nucleotide only (129 and 130 bp) have ΔLk values of -1.57 and -1.09. The last fragment on page 1 starting in position 111987 and the first one on page 2 differ

in two nucleotides (158 and 160 bp) and their ΔLk are -2.30 and -1.22. At first sight, this discrepancy questions the consistency of the method to estimate the ΔLk and it should be explained how these large differences between virtually identical fragments affects the reliability of the much smaller differences detected between nucleosomal DNA from different genomic positions.

Point 5.

As suggested, the authors have tested the ΔLk of individual fragments of nucleosomal DNA of different genomic origin and the results show a correlation with the mean ΔLk of each category (Figure 5d). These differences range between -1.2 and -1.36 (except in telomeric nucleosomes) and contrast with large differences between some nucleosomal DNA fragments that differ only in one or a few nucleotides (see previous point).

Point 6.

The lack of replicates (also requested by Reviewer #2) is not explicitly addressed by the authors on the grounds of the consistency of the ΔLk under different degrees of digestion, experimental conditions and variation in the cutting of the two gel fractions. The authors should explain why they do not refer to the two replicates deposited in GEO.

Reviewer #2:

Remarks to the Author:

The authors have addressed my comments.

Reviewer #3:

Remarks to the Author:

The authors in their response to reviewers convincingly addressed all my concerns. At this point I have no objections and support publishing the manuscript in the Nature Communications.

RESPONSE TO REVIEWER COMMENTS

We are very grateful to the three reviewers for their careful work and recognition of our previous round of review. We are pleased that Reviewers #2 and #3 have no further objections and support the publication of the manuscript. We also appreciate the remaining issues raised and the additional controls suggested by reviewer #1, which allowed us to strengthen the conclusions of the study. Our detailed responses to the reviewer's comments are below.

REVIEWER COMMENTS

Reviewer #1 (Remarks to the Author):

Point 1.

I suggested that sequenced mononucleosomal DNA should be mapped onto the reference genome to generate a genome-wide nucleosomal occupancy map to be compared to previously reported maps. Alternatively, mapping could have been done with nucleosomal DNA represented in the library to assess its quality relative to previous studies. This is a very straightforward control that does not exclude the control that the authors did by overlapping the sequences of the library to a catalog of nucleosome positions previously reported (new Suppl Fig 2a). An important negative control missing in that figure is the overlap to the same intervals of the same number of fragments 150 bp long selected at random bioinformatically in the *S. cerevisiae* genome. Results should be shown in histogram bars next to those of the library in the same figure for direct comparison of the enrichment of the library in nucleosomal DNA. In Suppl Fig 2b, the authors compare the overlap with random coordinates in the genome but, since most of the genome is complexed with nucleosomes, it is not clear how this comparison informs of the enrichment of the library in nucleosomal DNA. Again, as a negative control, the overlap ratio of the collection of 150 bp random fragments with the same random coordinates should be shown next to the random coordinates boxplot.

RESPONSE: We are aware that, in the previous round of revision, the reviewer had suggested mapping the sequenced mononucleosomal DNA library onto the reference genome. However, since our nucleosome library covered only 7 % of the yeast nucleosomes, we thought that such a low-density map would not be illustrative. Moreover, nucleosome occupancy profiles could not be generated because nucleosome identities were obtained from the limited number of yeast colonies hosting the library (one nucleosome per colony minichromosome). Therefore, we focused on demonstrating that the library was representative of the main nucleosome classes (Figure 1b-d; Figure 4a-c) and provided indicators of nucleosomal DNA quality (Supplementary Fig 2). Nevertheless, we have now conducted the nucleosome mapping exercise that the reviewer suggested by uploading our dataset of nucleosome coordinates on the Yeast Genome Browser (<https://browse.yeastgenome.org/>) and comparing it to other public nucleosomal maps. As expected, the nucleosomes of our library are sparse but consistent with previously mapped

nucleosomes. In the new Supplementary Fig 3 of the revised ms, we provide some indicative snapshots of the particular chromosomal regions with a relatively high nucleosome density that allow for some comparison with three different published nucleosomal maps obtained by MNase digestion (Lee 2007; Mavrich 2008 and Hu 2014). Importantly, these different nucleosomal maps reflect, themselves, some inherent variability in nucleosomal coordinates, which is characteristic of the positioning variation that occurs naturally and throughout the experimental settings of different laboratories.

New Supplementary Fig 3

Supplementary Fig. 3. Comparison of the nucleosomal DNA library with previously reported nucleosome occupancy maps. The dataset of nucleosome coordinates (Supplementary Data 1) was uploaded to the Yeast Genome Browser (<https://browse.yeastgenome.org/>). The 3 snapshots show particular chromosomal regions with a relatively high density of the uploaded nucleosomes (green tracks). For comparison, the snapshots include three different published nucleosomal maps (Lee 2007; Hu 2014, Mavrich 2008) obtained by micrococcal nuclease digestion.

As the reviewer suggested, we have also included negative controls missing in Supplementary Fig 2. Namely, in Supplementary Fig 2a, we have extended the histograms to incorporate the overlaps calculated for a randomly selected surrogate set of coordinates (of equal size and lengths). For each overlap bin, we have calculated a percentage ratio test assuming normally distributed values. In Supplementary Fig 2b, in addition to comparing the overlap of our library with the reference catalogue of nucleosomes and with randomized coordinates in the genome, we also compared the overlap of a random library of coordinates with randomized coordinates in the genome and with the reference catalogue of nucleosomes. We thank the reviewer for proposing these controls that illustrate more clearly the enrichment of our library in nucleosomal DNA.

Revised Supplementary Fig 2

Supplementary Fig. 2. Correspondence of the nucleosome library coordinates with previously referenced nucleosomes.

a, Degree of overlap (number of bp) of the nucleosomal DNA library coordinates with those of the catalogue of Jiang and Pugh (2009), which compiled six sets of nucleosome maps generated from different laboratories using different technologies. As a negative control, the overlap of the same number of fragments with randomized coordinates is shown.

b, Comparison of the overlapping (number of bp) of the nucleosome library with the reference catalogue of nucleosomes, and with randomized coordinates in the genome. As negative controls, the overlap of a random library of coordinates with randomized coordinates in the genome and with the reference catalogue of nucleosomes are shown.

c, GC-content (%) of the nucleosomal DNA library and that of the full catalogue of nucleosomes.

Point 2.

Related to the previous point, I suggested negative controls to make sure that the observed differences in ΔLk between nucleosomal DNA from different genomic locations (which is the main point of the work) were specific of nucleosomal DNA. In the absence of this control, it cannot be guaranteed that the ΔLk properties they attribute to nucleosomal DNA are not also detectable in non nucleosomal DNA. The fact that the mean value of ΔLk of -1.26 is the same in DNA fragments that overlap >90% as in those overlapping only 33% (50 bp) to the reference catalogue of nucleosomes (new Suppl Fig 6) leaves open the possibility that the -1.26 value could not be an exclusive property of nucleosomal DNA. Therefore, it is essential to make a library of non nucleosomal DNA (~ 150 bp fragments of naked DNA, even of prokaryotic origin with similar base composition to *S. cerevisiae*, partially digested with MNase) to compare the mean distribution of ΔLk with that shown in the new Suppl Fig 6. The authors claim that such library would not exclude that some nuclear proteins or nucleosomes could bind to them. This is true, but it would be unlikely that the mean ΔLk of an entire non nucleosomal library, or of a representative sample of individual fragments, would be close to -1.26 if that value is specific of nucleosomal DNA. In their reply, the authors explain that prior to estimating the differences between ΔLk_{Chr} and $\Delta Lk_{Chr+nuc}$ they first subtract the Lk of the relaxed minichromosome from that of the chromatinized counterpart (Suppl Fig 3). That approach was used in a previous article (Segura et al. 2018) but not in this work where, unless I am missing something, the topoisomerase relaxed minichromosome library was not used as a reference. Instead, Lk was estimated from the ratio between the representation of the individual DNA fragments in two gel fractions (Figure 3).

RESPONSE: Certainly, as the reviewer remarks, one cannot entirely exclude that ΔLk differences between nucleosomal DNAs from different genomic locations could be unrelated to nucleosome formation. However, we believe this to be unlikely because it is hard to imagine other mechanisms than nucleosome formation with genome-wide capacity to restrain negative supercoils with ΔLk about -1.26. The partial overlap of some nucleosome DNAs with the reference catalogue does not mean that these DNAs cannot be nucleosomal, since the reference catalogue is itself a compilation of variable positions. We appreciate the control experiment suggested by the reviewer, although it is quite laborious and nucleosomes might still form in non-nucleosomal DNAs. However, we agree with the reviewer in that it would be unlikely that the ΔLk restrained by non-nucleosomal DNAs would be close to -1.26. Since this measurement has not been done in past studies of DNA topology, we considered it an interesting add-on to support the conclusions of our study. What we have done is digest the genomic DNA of *E. coli* with Mnase, isolate fragments of ≈ 150 bp, insert them into the YCp1.3 minichromosome and calculate the average ΔLk constrained by these non-nucleosomal DNAs, exactly as we did to calculate the average ΔLk constrained nucleosomal DNAs in Segura et al (2018). We found that the non-nucleosomal DNAs produced an average ΔLk of about -0.84, which demonstrates that prokaryotic DNA is less suited than eukaryotic DNA to assemble nucleosomes. This result further substantiates that the range of ΔLk^{nuc} values reported in our study is specific to nucleosomal DNA. We included this new experiment as Supplementary Fig. 5 in the revised manuscript and referred to it in the main text accordingly (page 5).

Regarding the question about why the topoisomerase relaxed DNA library was not used as a reference in our present study, this is so because we estimated the average ΔLk^{nuc} of the nucleosomal library by directly comparing its Lk distributions to those of individual constructs with ΔLk^{nuc} -1.26 (Figure 2a). However, in the new experiment that compared the nucleosomal and non-nucleosomal DNA libraries (Supplementary Fig. 5), we used the topoisomerase relaxed

minichromosome libraries as a reference, the same approach used in Segura et al. (2018). The results corroborate that the average ΔLk^{nuc} of the nucleosomal library is -1.26.

New Supplementary Fig. 5

Supplementary Fig. 5. Comparison of the ΔLk restrained by nucleosomal and non-nucleosomal DNA libraries. **a**, MNase digestion of E.coli genomic DNA and gel purification of DNA fragments of ≈ 150 bp in length (L). Marker DNAs (M). This collection of non-nucleosomal DNA fragments was processed and inserted in the Ycp1.3 minichromosome as described for the nucleosomal DNA library. **b**, Gel electrophoresis of the Lk distributions of the Ycp1.3 minichromosome after adding one individual nucleosome of $\Delta Lk^{nuc} -1.26$ (lanes 1 and 2), the nucleosomal DNA library (lanes 3 and 4), and the non-nucleosomal DNA library (lanes 5 and 6). As explained in Supplementary Fig. 4, the Lk distribution constrained by the minichromosomes (Chr) (lanes 2, 4 and 6) is compared to that of the relaxed DNAs (Rel) (Lanes 1, 3, and 5). **c**, Plots to calculate the DNA linking number difference ($\Delta Lk^{Chr+nuc}$) constrained by the minichromosomes described above. As explained in Supplementary Fig. 4, $\Delta Lk^{Chr+nuc}$ is the distance in Lk units from Lk^{Chr+nuc} (Chr) to Lk⁰ (Rel). The value of ΔLk^{nuc} (or the inserted element) equals Lk^{Chr+nuc} - 5.81, which is the Lk difference constrained by the empty Ycp1.3 minichromosome. In agreement with our previous study (Segura *et al* 2018), the nucleosomal DNA library produced an average ΔLk^{nuc} of about -1.26. The non-nucleosomal DNA library produced an average ΔLk^{nuc} of about -0.84, which demonstrated that prokaryotic DNA is less suited than eukaryotic DNA to assemble nucleosomes.

Point 3.

The authors have shown that variations in the size of linker DNA do not affect ΔLk (Suppl Fig 7). Unlike in the analysis of the entire library (Figure 3), in this case, they estimate ΔLk using the ratio between the chromatinized and relaxed versions of the minichromosome.

RESPONSE: This is correct. To test if ΔLk^{nuc} would vary with the size of linker DNA, the ΔLk^{nuc} of different constructs was measured by subtracting ΔLk^{Chr} (-5.81) from their $\Delta Lk^{Chr+nuc}$. As shown in Supplementary Fig 9 (old Supplementary Fig 7), the value of $\Delta Lk^{Chr+nuc}$ was measured as the difference (not the ratio) between the chromatinized and relaxed versions of each minichromosome.

Point 4.

Since the authors show that the length of the DNA fragments inserted in the minichromosome does not alter the ΔLk after applying the 0.007 Lk unit/bp correction (Suppl Fig 7), how can it be explained that overlapping fragments that differ in one or a few bp generate a large difference in their ΔLk . Just to mention a couple of examples on page 1 (out of 86 pages) in Table 3, the two first fragments starting in position 26200 that differ in one nucleotide only (129 and 130 bp) have ΔLk values of -1.57 and -1.09. The last fragment on page 1 starting in position 111987 and the first one on page 2 differ in two nucleotides (158 and 160 bp) and their ΔLk are -2.30 and -1.22. At first sight, this discrepancy questions the consistency of the method to estimate the ΔLk and it should be explained how these large differences between virtually identical fragments affects the reliability of the much smaller differences detected between nucleosomal DNA from different genomic positions.

RESPONSE: The point that the reviewer is raising has not escaped our attention. Cases where a small positional offset in the coordinates of our nucleosomes is accompanied by a significant difference in the calculated ΔLk value do, indeed, exist. They are however rare. Indicatively, only 90 out of the 3404 overlapping pairs (2.6%) nucleosomes have ΔLk^{nuc} values that are >1.5-fold different (greater or smaller) than one of their neighbours and only 152 (4.4%) have values >1.3-fold. Such discrepancies are expected given: a) the inherent variability in nucleosome positioning that exists naturally in cellular populations and has been reported in the literature and b) the process of our calculations which builds upon many processing steps, each of which unavoidably incorporates some degree of noise in the final values. This is precisely why, throughout our study, we have primarily focused on the presentation, interpretation and discussion of mean values, which are (by definition) more representative of the populations and more descriptive of general trends that can be meaningfully associated with underlying molecular and biophysical properties of chromatin, instead of focusing on particular, point-like nucleosomes.

Point 5.

As suggested, the authors have tested the ΔLk of individual fragments of nucleosomal DNA of different genomic origin and the results show a correlation with the mean ΔLk of each category (Figure 5d). These differences range between -1.2 and -1.36 (except in telomeric nucleosomes) and contrast with large differences between some nucleosomal DNA fragments that differ only in one or a few nucleotides (see previous point).

RESPONSE: This is correct. Upon measuring the ΔLk^{nuc} of individual nucleosomal DNAs of different genomic origins (Supplementary Figure 12), the results show a good correlation with the mean ΔLk of each category calculated via Topo-seq. As indicated in the previous point, the interpretation and discussion of mean values are more representative of the populations and more descriptive of general trends.

Point 6.

The lack of replicates (also requested by Reviewer #2) is not explicitly addressed by the authors on the grounds of the consistency of the ΔLk under different degrees of digestion, experimental conditions and variation in the cutting of the two gel fractions. The authors should explain why they do not refer to the two replicates deposited in GEO.

RESPONSE: This is correct. We explained why the classification of the nucleosomes via Topo-seq should persist regardless of potential variability in the degree of chromatin digestion, experimental conditions, or experimental errors in cutting the gel. Our study analysed an intrinsic property of nucleosomal DNAs and, to do so, we compared the topology of all these DNAs in a common environment (the YCp1.3 minichromosome). The average ΔLk^{nuc} of different nucleosome classes revealed via topo-seq was then validated via the analysis of individual minichromosomes. The two replicates deposited in GEO are sequencing replicates. Namely, CNAG sequenced each sample in two Flowcell Lane Index units and we subsequently combined the sequences from both lanes to conduct the analyses. We thank the reviewer for noticing this point, which is now clarified in the methods section.

Reviewer #2 (Remarks to the Author):

The authors have addressed my comments.

We thank the reviewer for positively acknowledging our previous revision.

Reviewer #3 (Remarks to the Author):

The authors in the response to reviewers convincingly addressed all my concerns. At this point, I have no objections and support publishing the manuscript in Nature Communications.

We thank the reviewer for positively acknowledging our previous revision.

Reviewers' Comments:

Reviewer #1:

Remarks to the Author:

I acknowledge the effort that Segura et al. have done to address my concerns by adding new information and experimental evidence. I think that the data on the new Suppl Figures 3 and 5 and the new controls on Suppl Figure 2 provide stronger support for the nucleosomal origin of the fragments analysed and for the reliability of ΔL_{knuc} of -1.26 for nucleosomal DNA. I have no further comments and I support the publication of the manuscript in Nature Communications

However, I have two suggestions for the authors and the Editor to consider. First, the explanation for point 4 of my comments on the percentage of overlapping fragments showing differences in ΔL_{knuc} should be included in the legend of Table 2. Second, there are no biological replicates in this study.

This is very uncommon in genomic analysis. It is misleading to refer to the only dataset analysed in this study as Replicate 1 and Replicate 2 in the GEO repository, because they correspond to a single library sequenced in two lanes of the same flowcell. In fact, the reads from the two lanes were combined into a unique dataset for the analyses presented in the manuscript. This should be clearly stated in the methods section of the manuscript and in GEO.

Response to the reviewers' comments

Reviewer #1 (Remarks to the Author):

I acknowledge the effort that Segura et al. have done to address my concerns by adding new information and experimental evidence. I think that the data on the new Suppl Figures 3 and 5 and the new controls on Suppl Figure 2 provide stronger support for the nucleosomal origin of the fragments analysed and for the reliability of ΔLk_{nuc} of -1.26 for nucleosomal DNA. I have no further comments and I support the publication of the manuscript in Nature Communications

However, I have two suggestions for the authors and the Editor to consider. First, the explanation for point 4 of my comments on the percentage of overlapping fragments showing differences in ΔLk_{nuc} should be included in the legend of Table 2. Second, there are no biological replicates in this study. This is very uncommon in genomic analysis. It is misleading to refer to the only dataset analysed in this study as Replicate 1 and Replicate 2 in the GEO repository, because they correspond to a single library sequenced in two lanes of the same flowcell. In fact, the reads from the two lanes were combined into a unique dataset for the analyses presented in the manuscript. This should be clearly stated in the methods section of the manuscript and in GEO.

RESPONSE

We are very grateful to reviewer #1 for appreciating our previous round of revision, for the new suggestions and for supporting the publication of the manuscript in Nature Communications.

As suggested, we have revised the legend of Table 2 (Supplementary Data 2) by adding the following:

"Note that, in some cases, a small positional offset in the nucleosome coordinates is accompanied by a significant difference in the obtained ΔLk_{nuc} value. They are however rare. Indicatively, only 90 out of the 3404 overlapping pairs (2.6%) nucleosomes have ΔLk_{nuc} values that are >1.5-fold different (greater or smaller) than one of their neighbours and only 152 (4.4%) have values >1.3-fold. Such discrepancies are expected given: a) the inherent variability in nucleosome positioning that exists naturally in cellular populations and b) the process of our calculation steps, which unavoidably incorporates some degree of noise in the final values. This is precisely why this study primarily focused on the presentation, interpretation and discussion of mean values, which are by definition more representative of the populations and more descriptive of general trends that can be meaningfully associated with underlying molecular and biophysical properties of chromatin".

As suggested, we have clarified in the methods section and the GEO repository, that replicates 1 and 2 correspond to a single library sequenced in two lanes of the same flowcell:

"The reads from two Flowcell lanes (sequencing replicates 1 and 2) were combined into a unique analysis dataset".